# Federated Robustness Propagation: Sharing Adversarial Robustness in Federated Learning

## Abstract

Federated learning (FL) emerges as a popular distributed learning schema that learns a model from a set of participating users without requiring raw data to be shared. One major challenge of FL comes from heterogeneity in users, which may have distributionally different (or *non-iid*) data and varying computation resources. Just like in centralized learning, FL users also desire model robustness against malicious attackers at test time. Whereas adversarial training (AT) provides a sound solution for centralized learning, extending its usage for FL users has imposed significant challenges, as many users may have very limited training data as well as tight computational budgets, to afford the data-hungry and costly AT. In this paper, we study a novel learning setting that propagates adversarial robustness from high-resource users that can afford AT, to those low-resource users that cannot afford it, during the FL process. We show that existing FL techniques cannot effectively propagate adversarial robustness among *non-iid* users, and propose a simple yet effective propagation approach that transfers robustness through carefully designed batch-normalization statistics. We demonstrate the rationality and effectiveness of our method through extensive experiments. Especially, the proposed method is shown to grant FL remarkable robustness even when only a small portion of users afford AT during learning.

## 1 Introduction

Federated learning (FL) (McMahan et al., 2017) is a learning paradigm that trains models from distributed users or participants (e.g., mobile devices) without requiring raw training data to be shared, alleviating the rising concern of privacy issues when learning with sensitive data and facilitating learning deep models by enlarging the amount of data to be used for training. In a typical FL algorithm, each user trains a model locally using their own data and a server iteratively aggregates users' incremental updates or intermediate models, converging to a model that fuses training information from all users. A major challenge in FL comes from the heterogeneity of users. One source of heterogeneity is distributional differences in training data collected by users from diverse user groups (Fallah et al., 2020; Zhu et al., 2021). Yet another source is the difference of computing resources, as different types of hardware used by users usually result in varying computation budgets. For example, consider an application scenario of FL from mobile phones (Hard et al., 2019), where different types of mobile phones (e.g., generations of the same brand) may have drastically different computational power.

Data heterogeneity should be carefully handled during the learning as a single model trained by FL may fail to accommodate the differences (Yu et al., 2020). A variety of approaches have been proposed to address the issue, such as customizing network structures (Li et al., 2020c; Arivazhagan et al., 2019) or tailoring training strategies (Fallah et al., 2020; Dinh et al., 2020) for each user. Even though hardware heterogeneity is ubiquitous, their impacts to FL processes and possible solutions have received very limited attention so far.

The impacts of the two types of heterogeneity become aggravated when participating users' desire adversarial robustness during the inference stage, against imperceptible noise that can significantly mislead model predictions. To address this issue, a straightforward extension of FL, federated

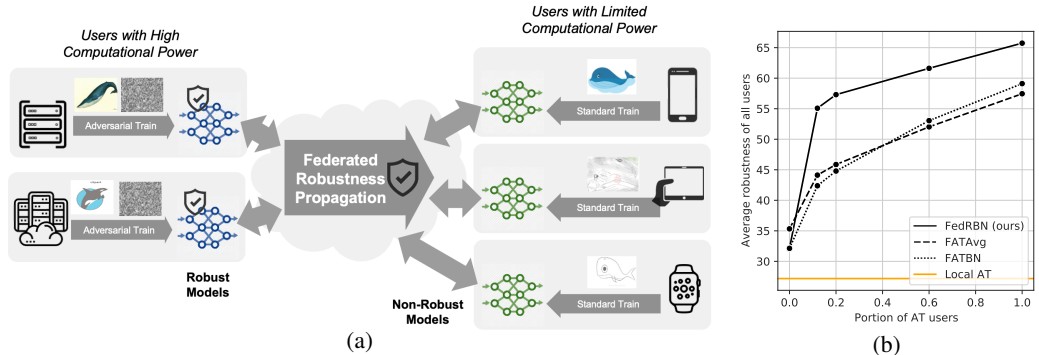

(a)             (b)

Figure 1: (a) We define a novel problem setting, where standard-training (ST) users, which might be limited by data or computational resources, can "share" robustness from adversarial-training (AT) users who can afford it. (b) Comparison of robustness on a varying portion of AT users, where a 5-domain digit recognition dataset is distributed to 50 users in total and details are in Appendix C.7.

adversarial training (FAT), can be adopted, which idea was explored in (Zizzo et al., 2020; Reisizadeh et al., 2020). Locally, each user trains models with adversarially augmented samples, namely adversarial training (AT) (Xie et al., 2020). As studied in central settings, the AT is data-thirsty and computationally expensive (Shafahi et al., 2019a). Therefore, involving a fair amount of users in FAT is essential, given the fact that each individual user may not have enough data to perform AT. However, this implies an increasing difficulty for fitting diverse data distributions and more intensive computation for each user, which could be $3 - 10$ times more costly than the standard equivalent (Shafahi et al., 2019a; Zhang et al., 2019). The computation overhead can be prohibitive for FL users with limited computational budget such as mobile devices. As such, it is often unrealistic to enforce *all* users in a FL process to conduct AT locally, despite the fact that the robustness is indeed a strongly desired or even required property for all users. This conflict raises a challenging yet interesting question: Is it possible to *propagate adversarial robustness in FL* so that budget-limited users can benefit from robustness training of users with abundant computational resources?

Motivated by the question above, we formulate a novel problem setting called Federated Robustness Propagation (FRP), as depicted in Fig. 1a. We consider a rather common non-iid FL setting that involves budget-sufficient users (AT users) that conduct adversarial training, and budget-limited ones (ST users) that can only afford standard training. The goal of FRP is to propagate the adversarial robustness from AT users to ST users. Note that sharing adversarial data is prohibited for mitigating the overhead of adversarial-data generation, due to the privacy consideration of the FL framework. In Fig. 1b, we show that independent AT by users without FL (`local AT`) will not yield a robust model since each user has scarce training data. Directly extending an existing FL algorithm *FedAvg* (McMahan et al., 2017) or a heterogeneity-mitigated one *FedBN* (Li et al., 2020c) with AT treatments, named as FATAvg and FATBN, give very limited capability of propagating robustness. The limitation may arise from the following two reasons: **1)** Heterogeneity of users results in distinct noise behavior in different users, degrading the transferability of model robustness. **2)** Structural differences between clean and adversarial samples may further impede robustness propagation (Xie & Yuille, 2019).

To address the aforementioned challenges, we propose a novel method Federated Robust Batch-Normalization (FedRBN) to facilitate propagation of adversarial robustness among FL users. **1)** We design a surprisingly simple linear method that transmits the robustness by copying batch-normalization (BN) statistics, inspired by the strong connection between model robustness and statistic parameters in the BN layer (Schneider et al., 2020); **2)** To efficiently propagate robustness among non-iid users, we weight and average multiple AT users' statistics as BN for every ST user; **3)** Facing the structural difference between the clean and adversarial data, we train two separate BNs for each data type, which are adaptively chosen at the inference stage. Our method is communication-efficient as it only incurs an one-time additional communication after training. We conduct extensive experiments demonstrating the feasibility and effectiveness of the proposed method. In Fig. 1b, we highlight some experimental results from Section 5. When only $20\%$ of non-iid users used AT during learning, the proposed FedRBN yields robustness, competitive with the best all-AT-user baseline (FATBN) by only a $2\%$ drop (out of $59\%$) on robust accuracy. Note that even if our method with $100\%$ AT users increase the upper bound of robustness, such a bound is usually not attainable because of the presence of resource-limited users that cannot afford AT during learning.

## 2 RELATED WORK

**Federated learning for robust models**. The importance of adversarial robustness in the context of federated learning, i.e., federated adversarial training (FAT), has been discussed in a series of recent literature (Zizzo et al., 2020; Reisizadeh et al., 2020; Kerkouche et al., 2020). Zizzo *et al.* (Zizzo et al., 2020) empirically evaluated the feasibility of practical FAT configurations (e.g., ratio of adversarial samples) augmenting FedAvg with AT but only in *iid* and label-wise non-*iid* scenarios. The adversarial attack in FAT was extended to a more general affine form, together with theoretical guarantees of distributional robustness (Reisizadeh et al., 2020). It was found that in a communication-constrained setting, a significant drop exists both in standard and robust accuracies, especially with non-*iid* data (Shah et al., 2021). In addition to the challenges investigated above, this work studies challenges imposed by hardware heterogeneity in FL, which was rarely discussed. Especially, when only limited users have devices that afford AT, we strive to efficiently share robustness among users, so that users without AT capabilities can also benefit from such robustness.

**Robust federated optimization**. Another line of related work focuses on the robust aggregation of federated user updates (Kerkouche et al., 2020; Fu et al., 2019). Especially, Byzantine-robust federated learning (Blanchard et al., 2017) aims to defend malicious users whose goal is to compromise training, e.g., by model poisoning (Bhagoji et al., 2018; Fang et al., 2020) or inserting model backdoor (Bagdasaryan et al., 2018). Various strategies aim to eliminate the malicious user updates during federated aggregation (Chen et al., 2017; Blanchard et al., 2017; Yin et al., 2018; Pillutla et al., 2020). However, most of them assume the normal users are from similar distributions with enough samples such that the malicious updates can be detected as outliers. Therefore, these strategies could be less effective on attacker detection given a finite dataset (Wu et al., 2020). Even though both the proposed FRP and Byzantine-robust studies work with robustness, they have fundamental differences: the proposed work focus on *the robustness during inference*, i.e., after the model is learned and deployed, whereas Byzantine-robust work focus on the robust learning process. As such, the proposed approach can combine with all Byzantine-robust techniques to provide training robustness.

## 3 PROBLEM SETTING: FEDERATED ROBUSTNESS PROPAGATION (FRP)

In this section, we will review AT, present the unique challenges from hardware heterogeneity in FL and formulate the problem of federated robustness propagation (FRP). In this paper, we assume that a dataset $D$ includes sampled pairs of images $x \in \mathbb{R}^d$ and labels $y \in \mathbb{R}^c$ from a distribution $\mathcal{D}$. Though we limit the data as images in this paper, our discussion could be generalized to other data forms. We model a classifier, mapping from the $\mathbb{R}^d$ data/input space to classification logits $f : \mathbb{R}^d \to \mathbb{R}^c$, by a deep neural network (DNN). Whenever not causing confusing, we use the symbol of a model and its parameters interchangeably. For brevity, we slightly abuse $\mathbb{E}[\cdot]$ for both empirical average and expectation and use $[N]$ to denote $\{1, \ldots, N\}$.

### 3.1 STANDARD TRAINING AND ADVERSARIAL TRAINING

An *adversarial attack* applies a bounded noise $\delta_\epsilon : \|\delta_\epsilon\| \leq \epsilon$ to an image $x$ such that the perturbed image $A_\epsilon(x) \triangleq x + \delta_\epsilon$ can mislead a well-trained model to give a wrong prediction. The norm $\|\cdot\|$ can take a variety of forms, e.g., $L_\infty$-norm for constraining the maximal pixel scale. A model $f$ is said to be *adversarially robust* if it can predict labels correctly on a perturbed dataset $\tilde{D} = \{(A_\epsilon(x), y) | (x, y) \in D\}$, and the standard accuracy on $D$ should not be greatly impacted.

Consider the following general learning objective:

$$\min_f L(f, D) = \min_f \frac{1}{|D|} \sum\nolimits_{(x,y) \in D} [(1-q)\, \ell_c(f; x, y) + q\, \ell_a(f; x, y)], \tag{1}$$

where $\ell_c$ is a standard classification loss on clean images and $\ell_a$ is an adversarial loss promoting robustness. Eq. (1) performs *standard training* if $q = 0$, and *adversarial training* if $q \in (0, 1]$. Without loss of generality, we limit our discussion for $q$ as 0 or 0.5. A popular instantiation of Eq. (1) is based on PGD attack (Madry et al., 2018; Tsipras et al., 2019): $\ell_c(f; x, y) = \ell(f(x), y)$, $\ell_a(f; x, y) = \max_{\|\delta\| \leq \epsilon} \ell(f(x+\delta), y)$, where $\|\cdot\|$ is the $L_\infty$-norm, $\ell$ can be the cross-entropy loss, i.e., $\ell(f(x), y) = -\sum_{t=1}^c y_t \log(f(x)_t)$ where $t$ is the class index and $f(x)_t$ represents the $t$-th output logit.

## 3.2 PROBLEM SETUP AND CHALLENGES

We start with a typical FL setting: a finite set of distributions $\mathcal{D}_i$ for $i \in [C]$, from which a set of datasets $\{D_k\}_{k=1}^K$ are sampled and distributed to $K$ users' devices. The users from distinct domains related with $\mathcal{D}_i$ expect to optimize objectives like Eq. (1). Some users can afford AT training (*AT users* from group $S$ with $q = 0.5$) when the remaining users cannot afford and use standard training (*ST users* from group $T$ with $q = 0$). If the two groups of users train models separately, the models of ST users will be much less robust than those of AT ones. Note that data exchange among users is forbidden according to the the FL setting for privacy concerns. The goal of *federated robustness propagation (FRP)* is to transfer the robustness from AT users to ST users at minimal computation and communication costs while preserve data locally. Formally, the FRP objective minimizes:

$$\mathrm{FRP}(\{f_k\}; \{D_k | D_k \sim \mathcal{D}_i\}) \triangleq \sum_{k \in T} \frac{1}{|D_k|} \sum_{(x,y) \in D_k} \ell_c(f_k)$$
$$+ \sum_{k \in S} \frac{1}{|D_k|} \sum_{(x,y) \in D_k} \frac{1}{2} [\ell_c(f_k) + \ell_a(f_k)]. \tag{2}$$

Note that different from FAT (Zizzo et al., 2020), FRP assumes that $D_k$ is sampled from different distributions and that there are at least one zero entry in $\boldsymbol{q}$. In the federated setting, each user's model is trained separately when initialized by a global model, and is aggregated to a global model at the end of each epoch. A popular aggregation technique is FedAvg (McMahan et al., 2017), which averages parameters by $f = \frac{1}{K} \sum_{k=1}^K a_k f_k$ with normalization coefficients $a_k$ proportional to $|D_k|$.

Remarkably, Eq. (2) formalizes two types of common user heterogeneity in FL. The first one is the *hardware heterogeneity* where users are divided into two groups of different computation budgets. A node of tight computation budget, e.g., smartphone, may join FL in group $T$, while a powerful one, e.g., desktop computer in $S$ (Hard et al., 2019). Besides, *data heterogeneity* is represented as $\mathcal{D}_i$ differing by $i$. We limit our discussion as the common feature distribution shift (on $x$) in contrast to the label distribution shift (on $y$), as previously considered in (Li et al., 2020c). Such distribution shift often happens when users are distributed across different environments, e.g., sensor data collected indoor and outdoor.

**New Challenges.** We emphasize that *jointly* addressing the two types of heterogeneity in Eq. (2) forms a new challenge, distinct from either of them considered exclusively. First, the scarcity of AT group $S$ worsens the data heterogeneity as additional distribution shift in the hidden representations from adversarial augmentation (Xie & Yuille, 2019). That means even if two users sample from the same distribution, their classification layers may operate on different distributions.

Second, the data heterogeneity makes the transfer of robustness non-trivial (Shafahi et al., 2019b). Hendrycks *et al.* discussed the transfer of models adversarially trained on multiple domains and massive samples (Hendrycks et al., 2019). In (Shafahi et al., 2019b), Shafahi *et al.*firstly studied the transferability of adversarial robustness from one data domain to another without the data-hungry problem. They proposed fine-tuning the robustness-sensitive layers in a neural network on a target domain. Distinguished from Shafahi *et al.*'s work, the FRP problem focuses on propagating robustness from multiple AT users to multiple ST users who have diverse distributions. Thus, fine-tuning all source models in ST users is often not possible due to prohibitive computation costs.

## 4 METHOD: FEDERATED ROBUST BATCH-NORMALIZATION (FEDRBN)

### 4.1 ROBUSTNESS PROPAGATION BY COPYING DEBIASED BN LAYERS

In centralized learning, an important observation is that robustness is highly correlated with the BN statistics (Xie & Yuille, 2019). We extend this investigation to the FL setting, where we assume all other parameters are shared besides BN layers. There are significant differences in BN parameters (mean and variance) between ST and AT users from the same domain, as shown in Fig. 2a. This observation indicates that directly using local BN statistics can hardly grant robustness to an ST user, and suggests a possible way to transfer robustness through leveraging the BN layers from AT users in ST users upon predicting possible adversarial input images. However, the distributions of users from distinct domains can be quite different (Joaquin Quiñonero-Candela et al., 2008), and therefore directly copying BN among users can suffer from the distribution shift by domains. This motivates us to develop a shift-aware debiasing method.

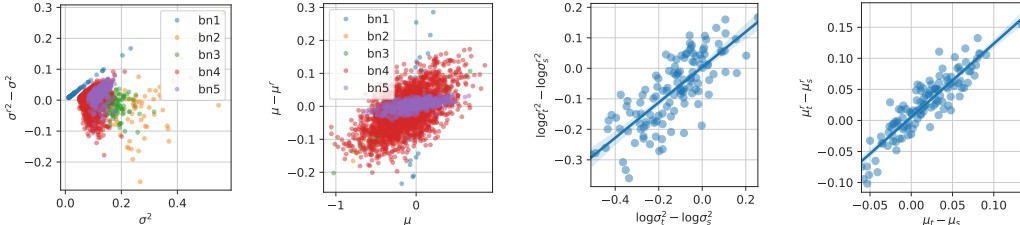

(a) Great difference of clean/noise statistics (y-axis)   (b) Correlation of relative statistics in the 2nd BN layer.

Figure 2: Models are trained with decoupled BN layers on Digits dataset. `bn1` is the first BN layer in the network. (a) Results on SVHN. (b) The relative statistics are compared on MNIST versus SVHN.

To capture the domain bias, we can leverage the BN layers as they are modeling local distributions. Ideally, differentially modeling two users will yield BN statistics for the two corresponding distributions, separately. However, one challenge here is that the BN statistics in non-iid AT and ST users are *biased simultaneously* by the domain difference and the adversarial noise. As such, directly differential modeling will capture the mixture of both types of bias and therefore would not be effective to infer the domain bias. Instead, we incrementally propose to simultaneously model clean data and noise data by BN for all AT users, since clean data are also available during training. To do so, we replace standard BN layers by ones that use the *dual batch-normalization (DBN)* structure (Xie et al., 2020), which keep two sets of BN statistics: one for clean data and one for noise data. The channel-wise mapping of DBN from the input layer $x$ to output $z$ is

$$z = w \left[ (1-h) \frac{x-\mu}{\sqrt{\sigma^2 + \epsilon_0}} + h \frac{x-\mu^r}{\sqrt{\sigma^{2^r} + \epsilon_0}} \right] + b, \tag{3}$$

where $\mu$ and $\sigma^2$ are the mean and variance over all non-channel dimensions, $\mu^r$ and $\sigma^{2^r}$ are their corresponding noised statistics, $h$ serves as a model-wise switch, which is $0$ for clean inputs or $1$ for noised inputs, $\epsilon_0$ is a small constant for the numerical stability. Different from prior work, e.g., (Xie et al., 2020; Xie & Yuille, 2019), we use a shared affine weights ($w$) and bias ($b$) for efficiency considerations. In the user-side training, we explicitly choose the clean or noise BN based on the input. Though we introduce DBN for a bias-inference purpose, the DBN still merits the model performance as it normalize representations more precisely as priorly investigated (Xie et al., 2020).

**Transfer robustness from a single user via copying BN layers**. With the clean BN embedded in DBN, we can estimate the distributional difference by an AT clean BN statistic tuple $(\mu_s, \sigma_s^2)$ and an ST (clean) BN statistic tuple $(\mu_t, \sigma_t^2)$. Formally, we propose a debiased statistic estimation by

$$\hat{\mu}_t^r = \mu_s^r + \lambda(\mu_t - \mu_s), \quad \hat{\sigma}_t^{r2} = \sigma_s^{r2} \left( \sigma_t^2 / (\sigma_s^2 + \epsilon_0) \right)^\lambda, \tag{4}$$

where $\lambda$ is a hyper-parameter in $[0, 1]$. Note that when the distributions are matched, i.e., $\mu_t = \mu_s$, then debiasing is not necessary and is automatically vanished. To justify the rationality of Eq. (4), we contrast the $\mu_s - \mu_t$ with $\mu_s^r - \mu_t^r$ in Fig. 2b (more results in Appendix D.3). The clean and noise BN statistics are estimated during training DBN, and we observe a strong correlation of the relative difference among domains both for the mean and variance.

To understand the debiasing method, we provide a principled analysis on a simplified one-dimensional example. We assume the noised inputs to a BN layer in user $s$ can be approximated by an affine-noise model $\tilde{x}_s = \lambda x_s + \nu$, $\tilde{x}_t = \lambda x_t + \nu$, where $x_s \sim \mathcal{N}(\mu_s, \sigma_s^2)$, $x_t \sim \mathcal{N}(\mu_t, \sigma_t^2)$ and $\nu \sim \mathcal{N}(\mu', \sigma'^2)$ is domain-independent noise. $\lambda$ is a constant scalar. We further assume $\nu$ is independent from $x_s$ and $x_t$. Taking expectation gives $\mu_s^r = \lambda \mu_s + \mu'$, $\mu_t^r = \lambda \mu_t + \mu'$; $\sigma_s^{r2} = \lambda^2 \sigma_s^2 + \sigma'^2$, $\sigma_t^{r2} = \lambda^2 \sigma_t^2 + \sigma'^2$. Due to the invariance assumption of $(\mu', \sigma')$, we have: $\hat{\mu}_t^r = \mu_s^r + \lambda(\mu_t - \mu_s)$, $\hat{\sigma}_t^r = \sqrt{\sigma_s^{r2} + \lambda^2 (\sigma_t^2 - \sigma_s^2)}$. However, $\hat{\sigma}_t^r$ is meaningless when $\sigma_s^{r2} + \lambda^2 (\sigma_t^2 - \sigma_s^2) < 0$. To fix this, we use a division instead of subtraction to represent the relative relation in Eq. (4). This simplified example by no means serves as rigorous analysis, which is an open problem outside the scope of this paper.

## 4.2 FEDRBN ALGORITHM AND ITS EFFICIENCY

We are now ready to present the proposed the two-stage Federated Robust Batch-Normalization (FedRBN) algorithm. During training (Algorithm 1), we train models locally with decoupled clean and noise BNs for each user. Locally trained models excluding BN are then aggregated by federated

parameter averaging. For ST users, they are free from heavy adversarial training. After training (Algorithm 2), the server aggregates BN statistic debiasing from AT users which are broadcasted to ST users for BN statistic estimation. For simplicity, we use $\mu$ and $\sigma^2$ for parameters in all layers.

---

**Algorithm 1** FedRBN: user training

**Input:** An initial model $f$ from the server, adversary $A(\cdot)$, dataset $D$, user budget type (AT or ST)
1: **for** mini-batch $\{(x, y)\}$ in $D$ **do**
2:     Set $f$ to use clean BN by $h \leftarrow 0$
3:     $L \leftarrow \mathbb{E}_{(x,y)}[\ell(f, (x, y))]$
4:     **if** user budget type is AT **then**
5:         Perturb data $\tilde{x} \leftarrow A(x)$
6:         Set $f$ to use noise BN by $h \leftarrow 1$
7:         $L \leftarrow \frac{1}{2} \left\{ L + \mathbb{E}_{(\tilde{x},y)}[\ell(f, (\tilde{x}, y))] \right\}$
8:     Update $f$ by one-step gradient descent
9: **Upload** parameters of layers except BN layers

---

**Algorithm 2** FedRBN: post-training

**Input:** AT users $S = \{s\}$ and ST user $T = \{t\}$
1: **for** AT users $s$ in $S$ locally **do**
2:     $\Delta \mu_s^r = \mu_s^r - \lambda \mu_s$
3:     $\Delta \log \sigma_s^{r2} = \log \sigma_s^{r2} - \lambda \log \sigma_s^2$
4:     Upload $\Delta \mu_s^r, \Delta \log \sigma_s^{r2}$ and to server
5: $\overline{\Delta \mu^r} = \frac{1}{|S|} \sum_{s \in S} \Delta \mu_s^r$
6: $\overline{\Delta \log \sigma^{r2}} = \frac{1}{|S|} \sum_{s \in S} \Delta \log \sigma_s^{r2}$
7: **for** ST users $t$ in $T$ locally **do**
8:     Download $\overline{\Delta \mu^r}, \overline{\Delta \log \sigma^{r2}}$
9:     $\hat{\mu}_t^r = \lambda \mu_t + \overline{\Delta \mu^r}$
10:    $\hat{\sigma}_t^{r2} = \exp \left[ \lambda \log \sigma_t^2 + \overline{\Delta \log \sigma^{r2}} \right]$

---

**Inference-stage BN selection (parameter $h$).** One issue of FedRBN is the choice of BN at inference time. To balance accuracy and robustness, we need a strategy to automatically select the right BN for each sample, i.e., applying noise BN when the input is found to be an adversarial one. Although the differences between clean data and adversarial are subtle in raw images, recent advances have shown promising results with the help of network structures. For example, in (Pang et al., 2018), the authors find that the non-maximal entropy of their logits are quite distinct. In (Feinman et al., 2017), their representations are separable under the similarity measurement of RBF-kernel (or K-density). Formally, we first compute the logits by $l = f(x)$ with $h \leftarrow 0$. Then we use $g(l)$ to predict whether the image is noised. If $g(l)$ indicates the image is noised, we assign $h \leftarrow g(l)$ and re-predict the label by $f(x)$. The details of locally training noise detectors is summarized in Algorithm 3. As only logits are required as input for $g(\cdot)$, the training is efficient. In Figs. 10 and 11, we visually show that adversarial examples can be identified by a classifier $g(\cdot)$ in the space of logits. More results are provided in Appendix C. Our strategy is similar to the layer-wise gating Liu et al. (2020), but we only use a gate function (namely noise detector) on model outputs, which can be efficiently trained without repeated adversarial-sample generation.

**Efficiency and privacy of BN operations**. Since the BN statistics are only a small portion of any networks and do not require back-propagation, an additional BN statistic will not significantly affect the efficiency (Wang et al., 2020). During training, because users do not send out BN layers, the communication cost is the same as a non-iid FL method (FedBN (Li et al., 2020c)) and less than other fully-shared methods like FedAvg (McMahan et al., 2017). After the training is done, BN layers will be copied to transfer robustness, and such one-time cost of transferring marginal components of networks will be neglectable, compared to the total cost incurred in FL. Noticeably, AT users will share the statistic difference ($\Delta \mu$) instead of local statistics ($\mu$) with the server, and the server will send the averaged parameters with ST users only. Especially when local statistics may leak users' data (Geiping et al., 2020), our method enjoys better privacy than traditional methods, like FedAvg.

## 5 EXPERIMENTS

**Datasets and models**. To implement a non-iid scenario, we adopt a close-to-reality setting where users' datasets are sampled from different distributions. We used two multi-domain datasets for the setting. The first is a subset (30%) of DIGITS, a benchmark for domain adaption (Peng et al., 2019b). DIGITS has $28 \times 28$ images and serves as a commonly used benchmark for FL (Caldas et al., 2019; McMahan et al., 2017; Li et al., 2020a). DIGITS includes 5 different domains: MNIST (MM) (Lecun et al., 1998), SVHN (SV) (Netzer et al., 2011), USPS (US) (Hull, 1994), SynthDigits (SY) (Ganin & Lempitsky, 2015), and MNIST-M (MM) (Ganin & Lempitsky, 2015). The second dataset is DOMAINNET (Peng et al., 2019a) processed by (Li et al., 2020c), which contains 6 distinct domains of large-size $256 \times 256$ real-world images: Clipart (C), Infograph (I), Painting (P), Quickdraw (Q), Real (R), Sketch (S). For DIGITS, we use a convolutional network with BN (or DBN) layers following each conv or linear layers. For the large-sized DOMAINNET, we use AlexNet (Krizhevsky et al., 2012) extended with BN layers after each convolutional or linear layer (Li et al., 2020c). One more large-sized image dataset is presented in Appendix C.5.

**Training and evaluation**. For AT users, we use $n$-step PGD (projected gradient descent) attack (Madry et al., 2018) with a constant noise magnitude $\epsilon$. Following (Madry et al., 2018), we use $\epsilon = 8/255$, $n = 7$, and attack inner-loop step size $2/255$, for training, validation, and test. We uniformly split the dataset for each domain into 10 subsets for DIGITS and 5 for DOMAINNET, following (Li et al., 2020c), which are distributed to different users, respectively. Accordingly, we have 50 users for DIGITS and 30 for DOMAINNET. Each user trains local model for one epoch per communication round. We evaluate the federated performance by standard accuracy (SA), classification accuracy on the clean test set, and robust accuracy (RA), classification accuracy on adversarial images perturbed from the original test set. All metric values are averaged over users. We defer other details of experimental setup such as hyper-parameters to Appendix C, and focus on discussing the results.

## 5.1 COMPREHENSIVE STUDY

To further understand the role of each component in FedRBN, we conduct a comprehensive study on its properties. In experiments, we use three representative federated baselines combined with AT: FedAvg (McMahan et al., 2017), FedProx (Li et al., 2020a), and FedBN (Li et al., 2020c). We use FATAvg to denote the AT-augmented FedAvg, and similarly FATProx and FATBN. To implement hardware heterogeneity, we let 20%-per-domain users from 3/5 domains (of DIGITS) conduct AT.

**Ablation Study**. We use FATBN as the base method and incrementally add FedRBN components: `+DBN`, `+detector` (for adaptively BN selection), `+copy` BN statistics, and `+debias` copying. Table 1 shows that even though DBN is a critical structure, simply adding DBN does not help unless the noise detector is applied. Also, the most influential component is copying, supporting our key idea. The `+debias` is more important in single AT domain case where domain gaps varies by different ST domains. Other than `+copy`, all other components barely affect the SA.

Table 1: Ablation of different FedRBN components. Standard deviations are enclosed in brackets.

| AT users | metric | base | +DBN | +detector | +copy | +debias |
|---|---|---|---|---|---|---|
| 20% in 5/5 domains | RA | 41.2 (1.3) | 38.8 (1.4) | 42.4 (1.4) | 55.7 (1.0) | 55.7 (1.3) |
| | increment | | -2.4 | +3.6 | **+13.3** | +0.0 |
| | SA | 86.4 (0.4) | 86.5 (0.4) | 86.2 (0.4) | 85.2 (0.6) | 85.2 (0.4) |
| 100% in 1/5 domain | RA | 36.5 (1.8) | 34.3 (2.0) | 37.3 (1.8) | 48.1 (1.6) | 49.8 (1.6) |
| | increment | | -2.2 | +3.0 | **+10.8** | +1.7 |
| | SA | 86.4 (0.4) | 86.4 (0.4) | 86.3 (0.4) | 84.3 (0.5) | 84.4 (0.4) |

**Impacts from Data Heterogeneity**. To study the influence of different AT domains, we set up an experiment where AT users only reside on one single domain. For simplicity, we let each domain contains a single user as in (Li et al., 2020c) and utilize only 10% of DIGITS dataset. The single AT domain plays the central role in gaining robustness from adversarial augmentation and propagating to other domains. The task is hardened by the non-singleton of gaps between the AT domain and multiple ST domains and a lack of the knowledge of domain relations. Results in Fig. 3a show the superiority of the proposed FedRBN, which improves RA for more than 10% in all cases with small drops in SA. We see that RA is the worst when MNIST serves as the AT domain, whereas RA propagates better when the AT domain is SVHN or SynthDigits. A possible explanation is that SVHN and SynthDigits are more visually different from the rest domains (see Fig. 6), forming larger domain gaps at test.

**Impacts from Hardware Heterogeneity**. We vary the number of AT users in training from $1/N$ (most heterogeneous) to $N/N$ (homogeneous) to compare the robustness gain. Fig. 3b shows that our method consistently improves the robustness. Even when all domains are noised, FedRBN is the best due to the use of DBN. When not all domains are AT, our method only needs half of the users to be noised such that the RA is close to the upper bound (fully noised case).

Other comprehensive studies including the parameter sensitivity and convergence are delayed to Appendix C.2. The impact of total data sizes is included in Appendix C.3.

## 5.2 COMPARISON TO BASELINES

To demonstrate the effectiveness of the proposed FedRBN, we compare it with baselines on two benchmarks. We repeat each experiment for three times with different seeds. We introduce two more baselines: personalized meta-FL extended with FAT (FATMeta) (Fallah et al., 2020) and federated

Table 2: Benchmarks of robustness propagation, where we measure the computation time ($T$) by counting $\times 10^{12}$ times of float add-or-multiplication operations (FLOPs).

| | Digits | | | | | | | | | DomainNet | | | | | | | | |
| | All | | | 20% | | | MNIST | | | All | | | 20% | | | Real | | |
| AT users | | | | | | | | | | | | | | | | | | |
| Metrics | RA | SA | T | RA | SA | T | RA | SA | T | RA | SA | T | RA | SA | T | RA | SA | T |
|---|---|---|---|---|---|---|---|---|---|---|---|---|---|---|---|---|---|---|
| FedRBN (ours) | **66.7** | **87.3** | 2218 | **56.2** | 85.3 | 665 | **49.8** | 84.3 | 665 | **35.9** | 60.5 | 38490 | **24.2** | 61.5 | 11547 | **21.6** | 61.9 | 10425 |
| FATBN | 60.0 | **87.3** | 2211 | 41.2 | **86.4** | 663 | 36.5 | **86.4** | 663 | 35.2 | 60.2 | 38363 | 20.3 | **63.2** | 11509 | 15.7 | **64.7** | 10390 |
| FATAvg | 58.3 | 86.1 | 2211 | 42.6 | 84.6 | 663 | 38.4 | 84.1 | 663 | 24.6 | 47.4 | 38363 | 15.4 | 57.8 | 11509 | 10.7 | 57.9 | 10390 |
| FATProx | 58.5 | 86.3 | 2211 | 42.8 | 84.5 | 663 | 38.1 | 84.1 | 663 | 24.8 | 47.1 | 38363 | 14.5 | 57.3 | 11509 | 10.4 | 57.1 | 10390 |
| FATMeta | 43.6 | 71.6 | 2211 | 35.0 | 72.6 | 663 | 35.3 | 72.2 | 663 | 6.0 | 23.5 | 38363 | 0.0 | 37.2 | 11509 | 0.1 | 38.1 | 10390 |
| FedRob | 13.1 | 13.1 | 2211 | 20.6 | 59.3 | 1032 | 17.7 | 48.9 | 645 | - | - | - | - | - | - | - | - | - |

Table 3: Compare FedRBN versus efficient federated AT on Digits.

| | 20% 3/5 AT domains | | 100% Free AT (Shafahi et al., 2019a) | |
| | FedRBN | FATAvg | FATAvg | FATBN |
|---|---|---|---|---|
| RA | **56.1** | 44.9 | 47.1 | 46.3 |
| SA | **86.2** | 85.6 | 63.6 | 57.4 |
| T | 273 | **271** | 276 | 276 |

Table 4: Evaluation of RA with various attacks on Digits. $n$ and $\epsilon$ are the step number and the magnitude of attack.

| Attack ($n$, $\epsilon$) | PGD (20,16) | PGD (100,8) | MIA (20,16) | MIA (100,8) | AA (-, 8) | LSA (7, -) | SA - |
|---|---|---|---|---|---|---|---|
| FedRBN | **42.8** | **54.5** | **39.9** | **52.2** | **48.3** | **73.5** | 84.2 |
| FATBN | 28.6 | 41.6 | 27.0 | 39.7 | 31.0 | 64.0 | **84.6** |
| FATAvg | 31.5 | 43.4 | 30.0 | 41.5 | 32.9 | 63.3 | 84.2 |

robust training (FedRob) (Reisizadeh et al., 2020). Because FedRob requires a project matrix of the squared size of image and the matrix is up to $256^2 \times 256^2$ on DOMAINNET which does not fit into a common GPU, we exclude it from comparison. Given the same setting, we constrain the computation cost in the similar scale for cost-fair comparison. We evaluate methods on two FRP settings. **1) Propagate from a single domain**. In reality, a powerful computation center may join the FL with many other users, e.g., mobile devices. Therefore, the computation center is an ideal node for the computation-intensive AT. Due to limitations of data collection, the center may only have access to a single domain, resulting gaps to most other users. We evaluate how well the robustness can be propagated from the center to others. **2) Propagate from a few multi-domain AT users**. In this case, we assume that to reduce the total training time, ST users are exempted from the AT tasks in each domain. Thus, an ST user wants to gain robustness from other same-domain users, but the different-domain users may hinder the robustness due to the domain gaps in adversarial samples.

Table 2 shows that our method outperforms all baselines for all tasks, while it associates to only marginal overhead (for fitting noise detector). Importantly, we show that only 20% users are enough to achieve robustness comparable to the best fully-trained baseline. To fully evaluate the robustness, we experiment with more attack methods, including MIA (Dong et al., 2018), AutoAttack (AA) (Croce & Hein, 2020) and LSA (Narodytska & Kasiviswanathan, 2016). A strong score-based blackbox attacks such as Square Attack (Andriushchenko et al., 2020) (included in AA) can avoid the trip fake robustness due to obfuscated gradient Even evaluated by different attacks (see Table 4), our method still outperforms others. With the concern that the attacker may bypass the noise detector and lead to the trip of fake robustness (Athalye et al., 2018), we include joint-attack on noise detector and the model prediction in Appendix C.8. Though, the accuracies of noise detectors are degraded somehow, the overall RA or SA of our method are still outstanding.

**Compare to full efficient AT**. In Table 3, we show that when computation time is comparable, our method can achieve both better RA and SA than full-AT baselines. For results to be comparable, we train FedRBN for limited 150 epochs while Free AT for 300 epochs. Although Free AT improves the robustness compared to FATAvg, it also greatly sacrifices SA performance. Thanks to stable convergence and decoupled BN, FedRBN maintains both accurate and robust performance though the AT is not 'free' for a few users.

**More federated configurations**. We also evaluate our method against FedBN with different federated configurations of local epochs $E$ and batch size $B$. We constrain the parameters by $E \in \{1, 4, 8\}$ and $B \in \{10, 50, 100\}$. The 20% 3/5 domain FRP setting is adopted with DIGITS dataset. In Table 5 (a longer one in Table 14), the competition results are consistent that our method significantly promotes robustness over FedBN. We also observe that both our method and FedBN prefer a smaller batch size and fewer local epochs for better RA and SA. In addition, our method drops less RA when $E$ is large or batch size increases.

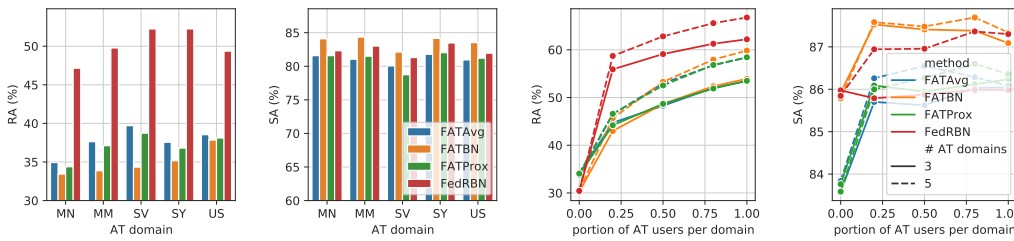

(a) FRP from a single AT domain      (b) FRP from partial AT users per domain

Figure 3: Evaluating FRP performance with different FRP settings.

**Evaluation with unequal dataset sizes.** Sub-sampling the same number of data points for each user may not be realistic in practice. Therefore, we study the experiment setting that sample sizes for users are different. The experiment details are given in Appendix C.1. In Table 6, we summarize the 3-repetition-averaged comparison results on the 20% 3/5 domain FRP setting on the DIGITS dataset. We see that our method is still most competitive with non-uniform dataset sizes.

Table 5: Evaluation with different FL configurations.

| local epochs $B$ | batch size $E$ | method | RA | SA |
|---|---|---|---|---|
| 10 | 1 | FedBN | 50.9 | **83.9** |
|  | 1 | FedRBN | **60.0** | 82.8 |
| 10 | 4 | FedBN | 42.0 | 75.8 |
|  | 4 | FedRBN | **56.3** | **76.1** |
| 50 | 1 | FedBN | 37.0 | **85.8** |
|  | 1 | FedRBN | **53.2** | 84.5 |

Table 6: Comparison with unequal user-dataset sizes.

|  | RA | SA |
|---|---|---|
| FedRBN (ours) | **53.1** | 84.4 |
| FedBN | 37.3 | **85.7** |
| FedAvg | 39.6 | 83.4 |
| FedProx | 39.5 | 83.4 |

**Domain-wise evaluation**. We note that performance in different domains could vary a lot. Simply comparing the performance averaged by users may not clearly present a fair comparison. As a consequence, we conduct experiments to compare different methods domain by domain. The basic settings follow Section 5.2. In Fig. 4a, one out of five domains is noised. Domains including SVHN, USPS, SynthDigits, MNIST-M are not augmented with adversarial samples. Therefore robustness is gained through federated propagation. Both in the easiest (USPS) and hardest (SVHN) cases, FedRBN outperforms baselines with higher RA and similar SA. In Fig. 4b, 20% users are noised in each domain. FedRBN improves the in-domain robustness propagation against FATBN by up to 20% (USPS). In summary, the propagation efficiency of FedRBN is consistent across different domains.

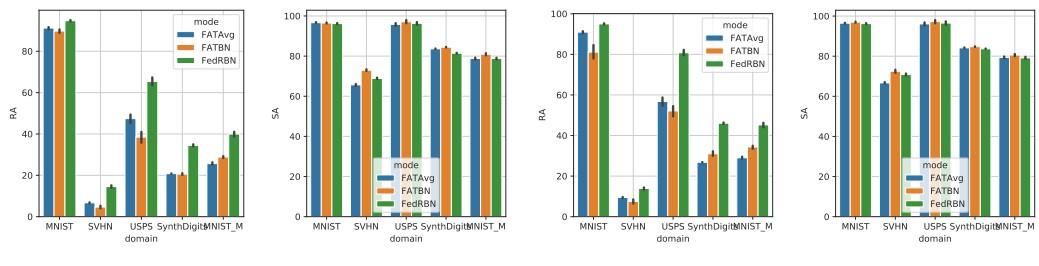

(a) Users from the MNIST domain are AT users      (b) 20% users are AT users

Figure 4: Comparison of robustness transfer approaches by domains.

## 6 CONCLUSION

In this paper, we investigate a novel problem setting, federate propagating robustness, and propose a FedRBN algorithm that transfers robustness in FL through robust BN statistics. Extensive experiments demonstrate the rationality and effectiveness of the proposed method, delivering both generalization and robustness in FL. We believe such a client-wise efficient robust learning can broaden the application scenarios of FL to users with diverse computation capabilities.

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

## A    ADDITIONAL RELATED WORK

This section reviews additional references in the areas of centralized adversarial learning and robustness transfer.

**Efficient centralized adversarial training**. A line of work has been motivated by similar concerns on the high time complexity of adversarial training. For example, Zhang *et al.* proposed to adversarially train only the first layer of a network which is shown to be more influential for robustness Zhang et al. (2019). Free AT Shafahi et al. (2019a) trades in some standard iterations (on different minibatches) for estimating a cached adversarial attack while keeping the total number of iterations unchanged. Wong *et al.* proposed to randomly initialize attacks multiple times, which can improve simpler attacks more efficiently Wong et al. (2019). Most of existing efforts above focus on speed up the local training by approximated attacks that trade in either RA or SA for efficiency. Instead, our method relocated the computation cost from budget-limited users to budget-sufficient users who can afford the expansive standard AT. As result, the computation expense is indeed exempted for the budget-limited users and their standard performance is not significantly influenced.

**Robustness transferring**. Our work is related to transferring robustness from multiple AT users to ST users. For example, a new user can enjoy the transferrable robustness of a large model trained on ImageNet Hendrycks et al. (2019). In order to improve the transferability, some researchers aim to align the gradients between domains by criticizing their distributional difference Chan et al. (2020). A similar idea was adopted for aligning the logits of adversarial samples between different domains Song et al. (2019). By fine-tuning a few layers of a network, Shafahi *et al.* shows that robustness can be transferred better than standard fine-tuning Shafahi et al. (2019b). Rather than a central algorithm gathering all data or pre-trained models, our work considers a distributed setting when samples or their corresponding gradients can not be shared for distribution alignment. Meanwhile, a large transferrable model is not preferred in the setting, because of the huge communication cost associating to transferring models between users. Because of the non-iid nature of users, it is also hard to pick a proper user model, that works well on all source users, for fine-tuning on a target user.

**Locally adapted models for data heterogeneity**. In the sense of modeling data heterogeneity, some prior work was done in adapting models for each local user (Smith et al., 2017; Arivazhagan et al., 2019; Fallah et al., 2020; Dinh et al., 2020). For example, Smith et al. (2017) studied the linear cases with regularization on the parameters, while we study a more general deep neural networks. In addition, the work did not consider a data-dependent adversarial regularization for better robustness, but a regularization that is independent from the data. Similarly, Dinh et al. (2020) regularizes the local parameters similar to the global model in $L_2$ distance, and Fallah et al. (2020) considers a meta-learning strategy instead. A simpler method was proposed by Arivazhagan et al. (2019) to only adapt the classifier head for different local tasks. Since all the above methods do not adapt the robustness from global to local settings, we first study how the robustness can be propagated among users in this work.

## B    ADDITIONAL TECHNICAL DETAILS OF FEDRBN

The structure of FedRBN is illustrated in Fig. 5. The DBN layer has the same input/output interface as an ordinary BN layer. Therefore, it can be plugged into most network structures whenever BN can be adopted. Except for DBN layers, FedRBN can be extended to other dual normalization layers, for instance, dual instance normalization Wang et al. (2020).

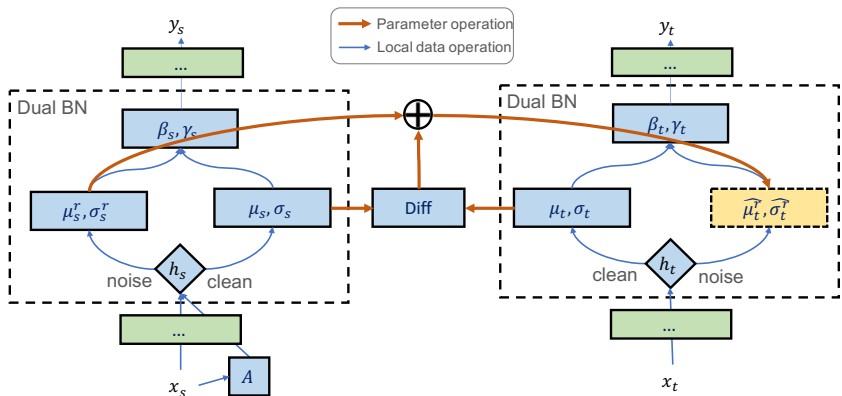

Figure 5: Illustration of the Dual-BN (DBN) layer and the copying operation for robustness propagation.

---

**Algorithm 3** Train an adversarial noise detector

**Input:** Adversary $A(\cdot)$, local validation dataset $D_{\text{val}}$
1: $D_a \leftarrow \emptyset$
2: Set $f$ to use clean BN by $h \leftarrow 0$
3: **for** mini-batch $\{(x, y)\}$ in $D_{\text{val}}$ **do**
4: $\quad \tilde{x} \leftarrow A(x)$
5: $\quad D_a \leftarrow D_a \cup \{(f(x), 0), (f(\tilde{x}), 1)\}$
6: Train a noise detector $g(\cdot)$ on $D_a$ by minimizing the cross-entropy loss:

$$g = \arg\min_g \mathbb{E}_{(l,y) \in D_a} \left[ \ell(g(l), y) \right] \quad (5)$$

7: **Return** noise detector $g(\cdot)$

---

**Multiple-source propagation by weighted BN averaging**. In FL with multiple AT users available, we suggest a strategy to transfer robustness from multiple sources. Given $N$ source BN statistics, we use a weighted average to estimate the noise BN of a target ST user $\hat{\mu}_t^r = \sum_i^N \alpha_i \hat{\mu}_{t,s_i}^r$, where $\hat{\mu}_{t,s_i}^r$ is the estimated value from user $s_i$ by Eq. (4). Likewise, $\hat{\sigma}_t^r$ can be estimated. However, the difference between the $s_i$-th adversarial distribution and the $t$-th counterpart is unknown. To tackle the issue, we first present the following result:

**Lemma B.1** (Informal). *Suppose the the divergence between any data distribution $\mathcal{D}$ and its adversarial distribution $\tilde{\mathcal{D}}$ is bounded, i.e., $d_{\mathcal{H}\Delta\mathcal{H}}(\tilde{\mathcal{D}}, \mathcal{D}) \leq d_\epsilon$ where $d_{\mathcal{H}\Delta\mathcal{H}}$ is $\mathcal{H}\Delta\mathcal{H}$-divergence in hypothesis space $\mathcal{H}$. If a target model is formed by $\alpha_i$-weighted average of models from $D_{s_i}$, the summation $\sum_i \alpha_i d_{\mathcal{H}\Delta\mathcal{H}}(D_{s_i}, D_t)$ of divergence between a set of source standard datasets $\{D_{s_i}\}$ and the target adversarial dataset $D_t$ weighted by $\alpha_i$ upper-bounds the generalization error of the target adversarial data distribution $\tilde{\mathcal{D}}_t$.*

The lemma extends an existing bound for federated domain adaptation (Peng et al., 2019b), and shows that the generalization error on the unseen target noised distribution $\tilde{D}_t$ is bounded by the $\alpha_i$-weighted distribution gaps. Motivated by the analysis, we set $\alpha_i$ to be reversely proportional to the divergence between $D_{s_i}$ and $D_t$. Specifically, we use a layer-averaged RBF-kernel to approximate the weight, i.e., $\alpha_i = \frac{1}{L} \sum_{l=1}^L \exp\left[-\gamma_{\text{rbf}} d_W^l(D_s, D_t)/p^l\right]$, where $p^l$ is the number of channels in layer $l$. The distribution discrepancy can be approximately measured by Wasserstein distance as $d_W^l(D_s, D_t) = \left\| \mu_{s_i}^l - \mu_t^l \right\|_2^2 + \left\| \sigma_{s_i}^l - \sigma_t^l \right\|_2^2$. We use a large $\gamma_{rbf}$, i.e., $100 \times \max_l p^l$, to contrast the in-distribution and out-distribution difference. Lastly, we normalize $\alpha_i$ such that $\sum_i \alpha_i = 1$. The formal analysis can be found in Appendix B.1.

As described in Algorithm 2, we propose to fit a support vector machine (SVM) (Cortes & Vapnik, 1995), denoted as $g(\cdot)$, with RBF kernels on the validation set of each user. At inference, we predict $h$ in Eq. (3) by $\hat{h} = g(f(x))$. The use of RBF kernels is partially inspired by (Feinman et al., 2017),

which used kernels on representations instead of logits. We use the most popular SVM instantiation, libsvm, to implement the detector which will automatically sparsify the support samples. Thus, it can successfully *avoid the over-parameterization* issue.

We are aware of efforts that eliminate the detected adversarial samples from test/inference, e.g., (Pang et al., 2018). However, simply refusing to predict suspicious samples may break up the downstream services, especially in the challenging scenarios when the detection becomes sensitive and many samples are suspicious. Instead, our method detects and predicts the suspicious samples using the robust model (with noise BN's), and does not refuse any predictions.

**The training and inference of noise detector**. Introducing a noise detector results in an additional training cost, but such overhead is marginal in practice. First, the training is only done once after the network training is done. We only need to forward the whole network once for each sample to obtain the logit required for training the noise detector. Therefore, the overall overhead for receiving robustness is extremely efficient as compared to the AT overhead. Suppose adversarial samples are collected at real time which could be private. Training a noise detector only requires approximately $a/T \times 100\%$ of the training adversarial samples where $T$ is the number of epochs and $a$ is the ratio of validation set versus the training set.

## B.1 PROOF OF LEMMA B.1

In this section, we use the notation $D$ for a dataset containing images and excluding labels. To provide supervisions, we define a ground-truth labeling function $g$ that returns the true labels given images. So as for distribution $\mathcal{D}$.

First, in Definition B.1, we define the $\mathcal{H}\Delta\mathcal{H}$-divergence that measures the discrepancy between two distributions. Because the $\mathcal{H}\Delta\mathcal{H}$-divergence measures differences based on possible hypotheses (e.g., models), it can help relating model parameter differences and distribution shift.

**Definition B.1.** Given a hypothesis space $\mathcal{H}$ for input space $\mathcal{X}$, the $\mathcal{H}$-divergence between two distributions $\mathcal{D}$ and $\mathcal{D}'$ is $d_{\mathcal{H}}(\mathcal{D}, \mathcal{D}') \triangleq 2\sup_{S \in \mathcal{S}_{\mathcal{H}}} |\Pr_{\mathcal{D}}(S) - \Pr_{\mathcal{D}'}(S)|$ where $\mathcal{S}_{\mathcal{H}}$ denotes the collection of subsets of $\mathcal{X}$ that are the support of some hypothesis in $\mathcal{H}$. The $\mathcal{H}\Delta\mathcal{H}$-divergence is defined on the symmetric difference space $\mathcal{H}\Delta\mathcal{H} \triangleq \{f(x) \oplus h'(x) | h, h' \in \mathcal{H}\}$ where $\oplus$ denotes the XOR operation.

Then, we introduce Assumption B.1 to bound the distribution differences caused by adversarial noise. The reason for introducing such an assumption is that the adversarial noise magnitude is bounded and the resulting adversarial distribution should not differ from the original one too much. Since all users are (or expected to be) noised by the same adversarial attacker $A_\epsilon(\cdot)$ during training, we can use $d_\epsilon$ to universally bound the adversarial distributional differences for all users.

**Assumption B.1.** *Let $d_\epsilon$ be a non-negative constant governed by the adversarial magnitude $\epsilon$. For a distribution $\mathcal{D}$, the divergence between $\mathcal{D}$ and its corresponding adversarial distribution $\tilde{\mathcal{D}} \triangleq \{A_\epsilon(x) | x \sim \mathcal{D}\}$ is bounded as $d_{\mathcal{H}\Delta\mathcal{H}}(\tilde{\mathcal{D}}, \mathcal{D}) \leq d_\epsilon$.*

Now, our goal is to analyze the *generalization error* of model $\tilde{f}_t$ on the target adversarial distribution $\tilde{\mathcal{D}}$, i.e., $L(\tilde{f}_t, \tilde{\mathcal{D}}_t) = \mathbb{E}_{\tilde{x} \sim \tilde{\mathcal{D}}_t}[|\tilde{f}_t(\tilde{x}) - g(\tilde{x})|]$. Since we estimate $\tilde{f}_t$ by a weighted average, i.e., $\sum_i \alpha_i \tilde{f}_{s_i}$ where $\tilde{f}_{s_i}$ is the robust model on $D_{s_i}$, we can adapt the generalization error bound from Peng et al. (2019b) for adversarial distributions. For consistency, we assume the AT users reside on the *source* clean/noised domains while ST users reside on the *target* clean/noised domains. Without loss of generality, we only consider one target domain and assume one user per domain.

**Theorem B.1** (Restated from Theorem 2 in Peng et al. (2019b)). *Let $\mathcal{H}$ be a hypothesis space of $VC$-dimension $d$ and $\{\tilde{D}_{s_i}\}_{i=1}^N$, $\tilde{D}_t$ be datasets induced by samples of size $m$ drawn from $\{\tilde{\mathcal{D}}_{s_i}\}_{i=1}^N$ and $\tilde{\mathcal{D}}_t$, respectively. Define the estimated hypothesis as $\tilde{f}_t \triangleq \sum_{i=1}^N \alpha_i \tilde{f}_{s_i}$. Then, $\forall \alpha \in \mathbb{R}_+^N$, $\sum_{i=1}^N \alpha_i = 1$, with probability at least $1 - p$ over the choice of samples, for each $f \in \mathcal{H}$,*

$$L(f, \tilde{\mathcal{D}}_t) \leq L(\tilde{f}_t, \tilde{\mathcal{D}}_s) + \sum_{i=1}^N \alpha_i \left( \frac{1}{2} d_{\mathcal{H}\Delta\mathcal{H}}(\tilde{\mathcal{D}}_{s_i}, \tilde{\mathcal{D}}_t) + \tilde{\xi}_i \right) + C, \tag{6}$$

where $C = 4\sqrt{\frac{2d\log(2Nm)+\log(4/p)}{Nm}}$, $\tilde{\xi}_i$ is the loss of the optimal hypothesis on the mixture of $\tilde{D}_{s_i}$ and $\tilde{D}_t$, and $\tilde{D}_s$ is the mixture of all source samples with size $Nm$. $d_{\mathcal{H}\Delta\mathcal{H}}(\hat{\mathcal{D}}_{s_i}, \tilde{\mathcal{D}}_t)$ denotes the divergence between domain $s_i$ and $t$.

Based on Theorem B.1, we may choose a weighting strategy by $\alpha_i \propto 1/d_{\mathcal{H}\Delta\mathcal{H}}(\tilde{\mathcal{D}}_{s_i}, \tilde{\mathcal{D}}_t)$. However, the divergence cannot be estimated due to the lack of the target adversarial distribution $\tilde{\mathcal{D}}_t$. Instead, we provide a bound by clean-distribution divergence in Lemma B.2.

**Lemma B.2** (Formal statement of Lemma B.1). *Suppose Assumption B.1 holds. Let $\mathcal{H}$ be a hypothesis space of $VC$-dimension $d$ and $\{D_{s_i}\}_{i=1}^N$, $D_t$ be datasets induced by samples of size $m$ drawn from $\{\mathcal{D}_{s_i}\}_{i=1}^N$ and $\mathcal{D}_t$. Let an estimated target (robust) model be $\tilde{f}_t = \sum_i \alpha_i \tilde{f}_{s_i}$ where $\tilde{f}_{s_i}$ is the robust model trained on $D_{s_i}$. Let $\tilde{D}_s$ be the mixture of source samples from $\{\tilde{D}_{s_i}\}_{i=1}^N$. Then, $\forall \alpha \in \mathbb{R}_+^N$, $\sum_{i=1}^N \alpha_i = 1$, with probability at least $1 - p$ over the choice of samples, for each $f \in \mathcal{H}$, the following inequality holds:*

$$L(f, \tilde{\mathcal{D}}_t) \leq L(\tilde{f}_t, \tilde{D}_s) + d_\epsilon + \sum_{i=1}^N \alpha_i \left( \frac{1}{2} d_{\mathcal{H}\Delta\mathcal{H}}(\mathcal{D}_{s_i}, \mathcal{D}_t) + \xi_i \right) + C,$$

*where $C$ and $\tilde{\xi}_i$ are defined in Theorem B.1. $D_s$ is the mixture of all source samples with size $Nm$. $d_{\mathcal{H}\Delta\mathcal{H}}(\mathcal{D}_{s_i}, \mathcal{D}_t)$ is the divergence over clean distributions.*

*Proof.* Notice that Eq. (6) is a loose bound as $d_{\mathcal{H}\Delta\mathcal{H}}(\tilde{\mathcal{D}}_{s_i}, \tilde{\mathcal{D}}_t)$ is neither bounded nor predictable. Differently, $d_{\mathcal{H}\Delta\mathcal{H}}(\mathcal{D}_{s_i}, \mathcal{D}_t)$ can be estimated by clean samples which is available for all users. Thus, we can bound $d_{\mathcal{H}\Delta\mathcal{H}}(\tilde{D}_{s_i}, \tilde{D}_t)$ with $d_{\mathcal{H}\Delta\mathcal{H}}(D_{s_i}, D_t)$. By Assumption B.1, it is easy to attain

$$\begin{aligned} d_{\mathcal{H}\Delta\mathcal{H}}(\tilde{D}_{s_i}, \tilde{D}_t) &\leq d_{\mathcal{H}\Delta\mathcal{H}}(\tilde{D}_{s_i}, D_{s_i}) + d_{\mathcal{H}\Delta\mathcal{H}}(D_{s_i}, D_t) + d_{\mathcal{H}\Delta\mathcal{H}}(D_t, \tilde{D}_t) \\ &\leq 2d_\epsilon + d_{\mathcal{H}\Delta\mathcal{H}}(D_{s_i}, D_t), \end{aligned} \tag{7}$$

where we used the triangle inequality in the space measured by $d_{\mathcal{H}\Delta\mathcal{H}}(\cdot, \cdot)$. Substitute Eq. (7) into Eq. (6), and we finish the proof. $\square$

In Lemma B.2, we discussed the bound for a $f \in \mathcal{H}$ (which also generalize to $\tilde{f}_t$) estimated by the linear combination of $\{\tilde{f}_{s_i}\}_i$. In our algorithm, $\tilde{f}_t$ and $\tilde{f}_{s_i}$ both represent the models with noise BN layers, and they only differ by the BN layers. Therefore, Lemma B.2 guides us to re-weight BN parameters according to the domain differences. Specifically, we should upweight BN statistics from user $s_i$ if $d_{\mathcal{H}\Delta\mathcal{H}}(\mathcal{D}_{s_i}, \mathcal{D}_t)$ is large, vice versa. Since $d_{\mathcal{H}\Delta\mathcal{H}}(\mathcal{D}_{s_i}, \mathcal{D}_t)$ is hard to estimate, we may use the divergence over empirical distributions, i.e., $d_{\mathcal{H}\Delta\mathcal{H}}(D_{s_i}, D_t)$ instead.

## C  ADDITIONAL EMPIRICAL STUDY RESULTS

In this section, we provide more details about our experiments and additional evaluation results.

### C.1  EXPERIMENT DETAILS

**Data**. By default, we use $30\%$ data of Digits for training. Datasets for all domains are truncated to the same size following the minimal one. In addition, we leave out $50\%$ ($60\%$ for DomainNet) of the training set for validation for Digits. Test sets are preset according to the benchmarks in Li et al. (2020c). Models are selected according to the validation accuracy. To be efficient, we validate robust users with RA while non-robust users with SA. We use a large ratio of the training set for validation, because the very limited sample size for each user will result in biased validation accuracy. When selecting a subset of domains for AT users, we select the first $n$ domains by the order: (MN, SV, US, SY, MM) for DIGITS, and (R, C, I, P, Q, S) for DOMAINNET. Some samples are plotted in Fig. 6 to show the visual difference between domains.

**Hyper-parameters**. We use a fixed parameters tuned by the DIGITS dataset and adopt it also for the DOMAINNET dataset: $\lambda = 0.5$, $C = 10$ and $\gamma = 1/10$. The same tuning strategy is applied for other baseline parameters.

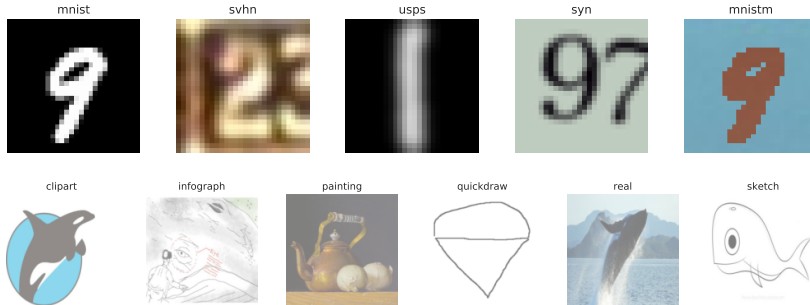

Figure 6: Visualization of samples.

Table 7: Network architecture for Digits dataset.

| Layer | Details |
|---|---|
| **feature extractor** | |
| conv1 | Conv2D(64, kernel size=5, stride=1, padding=2) |
| bn1 | DBN2D, RELU, MaxPool2D(kernel size=2, stride=2) |
| conv2 | Conv2D(64, kernel size=5, stride=1, padding=2) |
| bn2 | DBN2D, ReLU, MaxPool2D(kernel size=2, stride=2) |
| conv3 | Conv2D(128, kernel size=5, stride=1, padding=2) |
| bn3 | DBN2D, ReLU |
| **classifier** | |
| fc1 | FC(2048) |
| bn4 | DBN2D, ReLU |
| fc2 | FC(512) |
| bn5 | DBN1D, ReLU |
| fc3 | FC(10) |

**Network architectures** for DIGITS and DOMAINNET are listed in Tables 7 and 8. For the convolutional layer (Conv2D or Conv1D), the first argument is the number channel. For a fully connected layer (FC), we list the number of hidden units as the first argument.

**Training**. Following Li et al. (2020c), we conduct federated learning with 1 local epoch and batch size 32, which means users will train multiple iterations and communicate less frequently. Without specification, we let all users participant in the federated training at each round. Input images are resized to $256 \times 256$ for DOMAINNET and $28 \times 28$ for DIGITS. SGD (Stochastic Gradient Descent) is utilized to optimize models locally with a constant learning rate $10^{-2}$. Models are trained for 300 epochs by default. For FedMeta, we use the 0.001 learning rate for the meta-gradient descent and 0.02 for normal gradient descent following the published codes from Dinh et al. (2020). We fine-tune the parameters for DOMAINNET such that the model can converge fast. FedMeta converges slower than other methods, as it uses half of the batches to do the one-step meta-adaptation. We do not let FedMeta fully converge since we have to limit the total FLOPs for a fair comparison. FedRob fails to converge because locally estimated affine mapping is less stable with the large distribution discrepancy.

**Unequal users' data sample**. We assume a user samples a variable ratio of data, which follows a Dirichlet distribution. We plot the different sample sizes for users in Fig. 8. Due to the varying dataset sizes, we let each user run a fixed number of iterations which is calculated by the average number of the per-epoch iterations of all users.

We implement our algorithm and baselines by PyTorch. The FLOPs are computed by `thop` package in which the FLOPs of common network layers are predefined [1]. Then we compute the times of forwarding (inference) and backward (gradient computing) in training. Accordingly, we compute the total FLOPs of the algorithm. Because most other computation costs are relatively minor compared to the network forward/backward, these costs are ignored in our reported results.

---

[1]Retrieve the `thop` python package from https://github.com/Lyken17/pytorch-OpCounter.

Table 8: Network architecture for DomainNet dataset.

| Layer | Details |
|---|---|
| **feature extractor** | |
| conv1 | Conv2D(64, kernel size=11, stride=4, padding=2) |
| bn1 | DBN2D, ReLU, MaxPool2d(kernel size=3, stride=2) |
| conv2 | Conv2D(192, kernel size=5, stride=1, padding=2) |
| bn2 | DBN2D, ReLU, MaxPool2d(kernel size=3, stride=2) |
| conv3 | Conv2D(384, kernel size=3, stride=1, padding=1) |
| bn3 | DBN2D, ReLU |
| conv4 | Conv2D(256, kernel size=3, stride=1, padding=1) |
| bn4 | DBN2D, ReLU |
| conv5 | Conv2D(256, kernel size=3, stride=1, padding=1) |
| bn5 | DBN2D, ReLU, MaxPool2d(kernel size=3, stride=2) |
| avgpool | AdaptiveAvgPool2d(6, 6) |
| **classifier** | |
| fc1 | FC(4096) |
| bn6 | DBN1D, ReLU |
| fc2 | FC(4096) |
| bn7 | DBN1D, ReLU |
| fc3 | FC(10) |

Figure 7: The convergence curves and parameters sensitivity of $\lambda$, $C$ and $\gamma$. $C$ is for regularization and $\gamma$ is for RBF-kernel used in SVM whose performance is evaluated on Digits domains.

## C.2 CONVERGENCE AND HYPER-PARAMETERS

**Convergence**. The first plot in Fig. 7 shows convergence curves of different competing algorithms. Since FedRBN only differs from FATBN by a DBN structure, FATBN and FedRBN have similar convergence rates that are faster than others. We see that FedRBN converges even faster than FATBN. A possible reason is that DBN decouples the normal and adversarial samples, the representations after BN layers will be more consistently distributed among non-iid users.

**Parameter Sensitivity of the $\lambda$, $C$ and $\gamma$**. The second plot in Fig. 7 shows a preferred $\lambda$ is neither too small or too close to 1. Since $\lambda$ is critical when heterogeneity is severer, we evaluate the sensitivity as only one domain (MNIST in DIGITS) is adversarially trained. We find that a larger $\lambda$ is more helpful for the RA, as the estimation is closer to the true robust one. The rest plots in Fig. 7 demonstrate the stability of the noise detector when a choice of $C = 10$ and $\gamma = 1/10$ for SVM generally works.

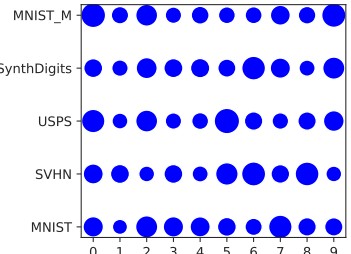

Figure 8: Dataset sizes for users when the global seed is set as 1. Larger circles indicate more training samples. The x-axis represents the user index.

## C.3 IMPACT OF DATA SIZE AND VALIDATION RATIO

To investigate the impact of data size, we conduct experiments with varying training dataset sizes and validation ratios. Experiments follow previous protocols on the Digits dataset. Following the training/testing split in Li et al. (2020c), we first sample a percentage of data for training. From the training set, we randomly select a subset for validation. We denote the two subsampling ratios as `training percentage` and `validation ratio`, respectively. When varying the training percentage, we fix the validation ratio at $10\%$. When varying the validation ratio, we use $30\%$

training data. As shown in Fig. 9a, more training samples can improve the robustness and our method outperforms baselines consistently. In Fig. 9b, the ratio of validation set is less influential for the robustness performance of our FedRBN, but a larger validation ratio can reduce the time complexity of training as less samples are used for gradient computation. Though baseline methods obtain higher robust accuracies with smaller validation ratios, our method still introduces large gains in all cases.

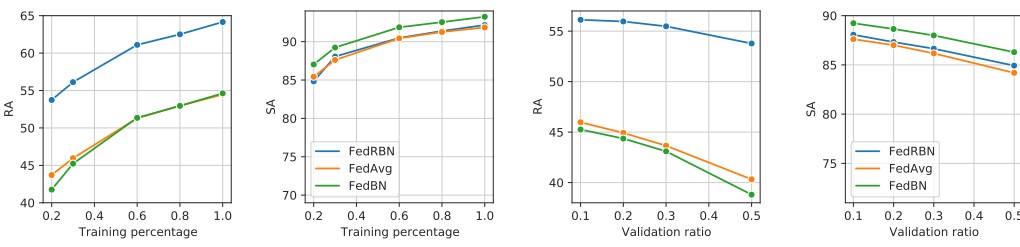

(a) Varying the size of training set.  (b) Varying the ratio of validation data in training set.

Figure 9: Experiments with varying data size.

## C.4 Logits of adversarial and clean examples

Visualization of logits by t-SNE is presented in Fig. 11 (DIGITS) and Fig. 10 (DOMAINNET). We generally observe that the clean and adversarial logits are separable with generalizable decision boundaries. Moreover, MNIST and USPS domains turn out to be the most separable cases as they are easier classification tasks compared to the rest ones. Though some domains have a few mixed samples in the visualization, their noise detection accuracies are mostly higher than 90%. Thus, it is rational to fit a noise detector on the validation set for helping BN selection at test time.

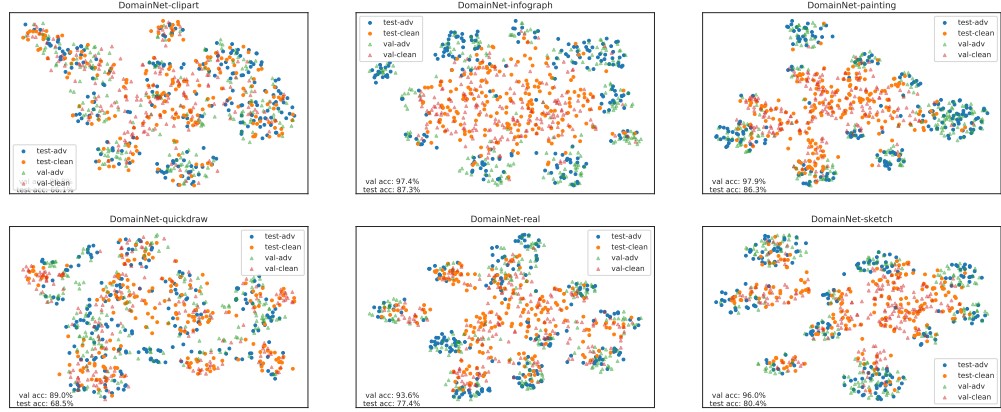

Figure 10: Logits of standard-trained models visualized by t-SNE on DOMAINNET.

## C.5 Results on the Office-Caltech10 Dataset

Following the same setting as DO-MAINNET experiments, we extend our experiments to a smaller dataset, Office-Caltech10 dataset preprocessed by Li et al. (2020c) with images acquired by different cameras. The dataset includes 4 domains: Amazon, Caltech, DSLR, Webcam. Because the dataset has very few samples, we only generate 2 users per domain such that each user has at least

Table 9: Comparison to baselines on the Office-Caltech10 dataset. Standard deviations are reported in brackets.

| AT users | Amazon | | All | |
|---|---|---|---|---|
| metric | RA | SA | RA | SA |
| FedRBN (ours) | **9.2 (3.4)** | 62.9 (3.4) | 29.1 (2.4) | **68.7 (1.7)** |
| FedBN | 5.1 (1.1) | **65.9 (2.4)** | **30.8 (2.5)** | 67.2 (2.1) |
| FedAvg | 0.6 (0.5) | 54.7 (3.8) | 13.3 (2.3) | 56.0 (2.6) |
| FedProx | 0.6 (0.6) | 55.3 (4.7) | 13.6 (1.8) | 56.2 (2.1) |

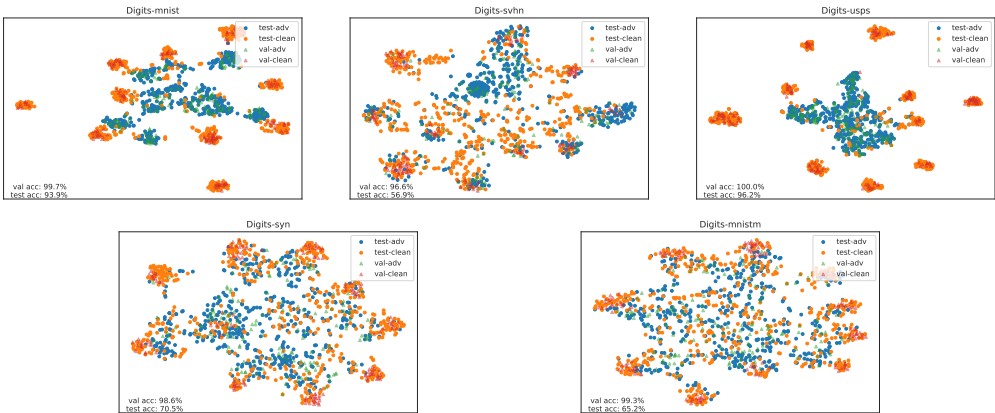

Figure 11: Logits of standard-trained models visualized by t-SNE on DIGITS.

100 samples. In Table 9, we see that our method outperforms baselines as only one domain is adversarially trained. As the training set is rather small, the RAs are generally worse than the ones on DIGITS or DOMAINNET.

Table 10: Comparison to robustness transferring by fine-tuning (FT).

|  | # FT iterations | # freeze layers | RA | SA |
|---|---|---|---|---|
| FedRBN | - | 0 | **53.1** | 84.4 |
| FedAvg | - | 0 | 44.7 | **85.7** |
| FedAvg+FT | 200 | 0 | 39.2 | 83.6 |
| FedAvg+FT | 200 | 3 | 31.6 | 78.2 |
| FedAvg+FT | 200 | 4 | 29.8 | 74.7 |
| FedAvg+FT | 200 | 5 | 31.5 | 66.1 |
| FedAvg+FT | 100 | 0 | 40.6 | 83.4 |
| FedAvg+FT | 100 | 3 | 32.0 | 77.5 |
| FedAvg+FT | 100 | 4 | 31.5 | 72.9 |
| FedAvg+FT | 100 | 5 | 31.5 | 64.5 |
| FedAvg+FT | 20 | 0 | 40.6 | 79.6 |
| FedAvg+FT | 20 | 3 | 33.4 | 73.8 |
| FedAvg+FT | 20 | 4 | 31.9 | 66.8 |
| FedAvg+FT | 20 | 5 | 31.9 | 62.2 |

### C.6 COMPARISON TO ROBUSTNESS TRANSFERRING BY FINE-TUNING

As an alternative to FRP, fine-tuning (FT) the federated-trained models on target users can enjoy even better efficiency than FedRBN. Here, we first train AT users by FedAvg for 300 epochs. Note that we do not adopt FedBN because FedBN will not output a single model for adapting to new users. Then, the model is used for initializing the models for ST users. These ST users will be trained by FedAvg for a given number of FT iterations. Still, we adopt the $20\%$ $3/5$ domain FRP setting on the DIGITS dataset. In Table 10, we see that such a fine-tuning does not improve the robustness (RA).

### C.7 EXPERIMENTS IN FIG. 1B

Though the results in Fig. 1b have been reported in previous experiments, we re-summarize the results in Table 11 for ease of reading. The basic setting follows the previous experiments on the Digits dataset. We construct different portions of AT users by *in-domain* or *out-domain* propagation settings. When robustness is propagated in domains, we sample AT users in each domain by the same portion and leave the rest as ST users. When robustness is propagated out of domains, all users from the last two domains will not be adversarially trained and gain robustness from other domains. Concretely, we add the FedRBN without copy propagation (FedRBN w/o prop) in the table, to show the propagation effect. FedRBN w/o prop outperforms the baselines only when the AT-user portion is more than 60%. Meanwhile, due to the lack of copy propagation, the RA is much worse

than the propagated `FedRBN`. Unless no AT user presents in the federated learning, FedRBN always outperforms baselines.

Table 11: Results and detailed configurations of Fig. 1b on the 5-domain Digits dataset. FedAvg and FedBN corresponds to FATAvg and FATBN in the figure.

| AT-user ratio | propagation | method | RA | SA | # AT domain | per-domain AT ratio |
|---|---|---|---|---|---|---|
| 0% | none | FedRBN (ours) | 32.1 | 84.3 | 0 | 0.0 |
| | | FedRBN w/o prop | 32.1 | 84.3 | 0 | 0.0 |
| | | FedAvg | **35.3** | 82.0 | 0 | 0.0 |
| | | FedBN | 32.1 | **84.3** | 0 | 0.0 |
| 12% | mix | FedRBN (ours) | **55.1** | 84.6 | 3 | 20% |
| | | FedRBN w/o prop | 38.7 | 84.5 | 3 | 20% |
| | | FedAvg | 44.1 | 84.1 | 3 | 20% |
| | | FedBN | 42.4 | **86.0** | 3 | 20% |
| 20% | in-domain | FedRBN (ours) | **57.3** | 85.3 | 5 | 20% |
| | | FedRBN w/o prop | 46.1 | **86.1** | 5 | 20% |
| | | FedAvg | 45.9 | 84.7 | 5 | 20% |
| | | FedBN | 44.8 | 86.0 | 5 | 20% |
| 60% | out-domain | FedRBN (ours) | **61.6** | 85.0 | 3 | 100% |
| | | FedRBN w/o prop | 56.2 | **85.5** | 3 | 100% |
| | | FedAvg | 52.0 | 84.2 | 3 | 100% |
| | | FedBN | 53.0 | **85.5** | 3 | 100% |
| 100% | none | FedRBN (ours) | 65.7 | **85.9** | 5 | 100% |
| | | FedRBN w/o prop | **65.8** | **85.9** | 5 | 100% |
| | | FedAvg | 57.5 | 84.7 | 5 | 100% |
| | | FedBN | 59.1 | **85.9** | 5 | 100% |

## C.8 JOINT ATTACK ON NOISE DETECTOR AND MODEL PREDICTION

Furthermore, to check if the adversarial attacker can bypass the noise detector, we use PGD to jointly maximize the sum of the cross-entropy losses of the noise detector and the inference model. We show the results on the Digits and DomainNet dataset following the settings of Table 2. For ease of comparison, we also include baseline results from Table 2. As observed in Table 12, our method FedRBN outperforms baselines consistently in all benchmarks.

To show the JointPGD does fool the noise detectors, we present the detection accuracy of adversary samples in Table 13. The detectors of FedRBN have lower accuracies due to the joint attack, compared to FedRBN under PGD attack.

Table 12: The robust accuracy of FedRBN under joint PGD attacks. Baseline results are present with PGD attacks. Standard deviations are included in brackets.

| | Digits | | | DomainNet | | |
|---|---|---|---|---|---|---|
| AT users | All | 20% | MNIST | All | 20% | Real |
| FedRBN (ours) | 61.4 (0.5) | 50.6 (1.3) | 41.8 (1.6) | 30.2 (0.3) | 23.7 (0.8) | 17.3 (1.2) |
| FATBN | 60.0 | 41.2 | 36.5 | 29.9 | 20.3 | 11.3 |
| FATAvg | 58.3 | 42.6 | 38.4 | 24.6 | 15.4 | 9.4 |
| FATProx | 58.5 | 42.6 | 38.1 | 24.8 | 14.5 | 9.4 |

Table 13: Detection accuracy by PGD and joint PGD attacks.

| Attack | Digits | | | DomainNet | | |
|---|---|---|---|---|---|---|
| | All | 20% | MNIST | All | 20% | Real |
| PGD | 79.2 | 81.4 | 79.5 | 56.2 | 63.8 | 65.8 |
| Joint PGD | 63.9 | 56.8 | 52.3 | 56.2 | 63.8 | 65.8 |

Table 14: Evaluation with different FL configurations

| $B$ | $E$ | method | RA | SA |
|---|---|---|---|---|
| 10 | 1 | FATBN | 50.9 | **83.9** |
| | 1 | FedRBN | **60.0** | 82.8 |
| 10 | 4 | FATBN | 42.0 | 75.8 |
| | 4 | FedRBN | **56.3** | **76.1** |
| 10 | 8 | FATBN | 30.9 | 63.1 |
| | 8 | FedRBN | **53.4** | **68.4** |
| 50 | 1 | FATBN | 37.0 | **85.8** |
| | 1 | FedRBN | **53.2** | 84.5 |
| 100 | 1 | FATBN | 35.7 | **85.3** |
| | 1 | FedRBN | **53.0** | 83.8 |

# D    ADDITIONAL EXPERIMENTS FOR REBUTTAL

## D.1    PARTIAL PARTICIPANTS

In reality, we can not expect that all users are available for training in each round. Therefore, it is important to evaluate the federated performance when only a few users can contribute to the learning. To simulate the scenario, we uniformly sample a number of users without replacement per communication round. Only these users will train and upload models. In Fig. 12, RA and SA are reported against the number of selected users. We observe that SA is barely affected by the partial involvement, while RA increases by fewer users per round. Since the actual update steps in the view of the global server are reduced with lower contact ratios, the result is consistent with Table 14, where smaller batch sizes or fewer local steps lead to better robustness.

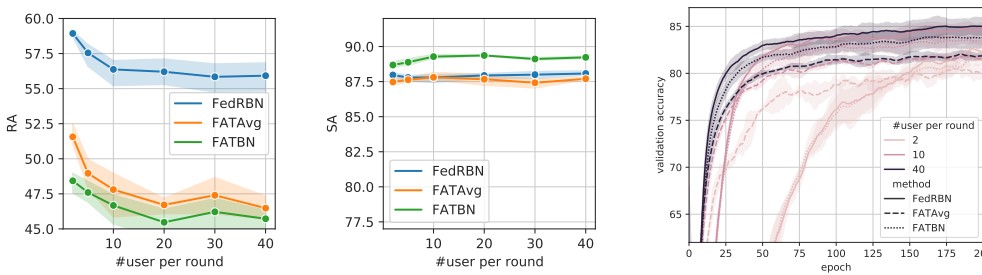

Figure 12: Vary the number of involved users per communication round. The validation accuracy is computed by averaging users' accuracy. For AT users, the RA is used while SA is used for ST users.

## D.2    SCALABILITY WITH MORE USERS

Since our method has the similar training/communication strategy as FATBN or FATAvg (except switching and copying BN which are quite lightweight), the federation of FedRBN and its complexity scale up to more users like FATBN or FATAvg who are widely used scalable implementations. To empirically evaluate the scalability of our method versus FATBN and FATAvg, we experiment with more clients given the Digits dataset. With the same total training samples, we re-distribute the data to different numbers of clients in a non-uniform manner. In Fig. 13, we evaluate the RA and SA by increasing the total number of users, including 25, 50, 150, 200. In each communication round, 50% randomly selected users will upload their trained models. The trend shows that both RA and SA will be lower when samples are distributed to more clients. Despite the degradation, our method maintains advantages in RA consistently.

In Fig. 13, we also demonstrate that our method converges faster than baselines either with fewer or more users. The validation accuracy is computed by averaging users' accuracy when RA is used for AT users and SA for ST users. As observed, when data are more concentrated in a few users (i.e. smaller numbers of users), the convergence will be faster. The result is natural for most non-iid federated learning problems. For example, Li et al. (2020b) proved that more clients will result in worse final losses and slower convergence.

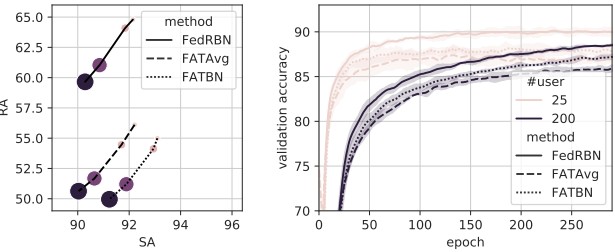

Figure 13: Robustness and accuracy by the increasing total number of users as 25, 50, 150, and 200. The larger scatter in the left figure indicates more users.

## D.3    MORE CORRELATION PLOTS OF BN STATISTICS

We provide more correlation plots of BN statistics in Fig. 14 as supplementary to Fig. 2b. In most layers, the cross-domain difference of BN statistics is linearly correlated between clean and robust

ones. The linear correlation is weak in some shallow layers. One reason is that the coupling of adversarial noise and data undermine the noise-independence assumption made for Eq. (4). The decoupling is further discussed in Appendix D.5.

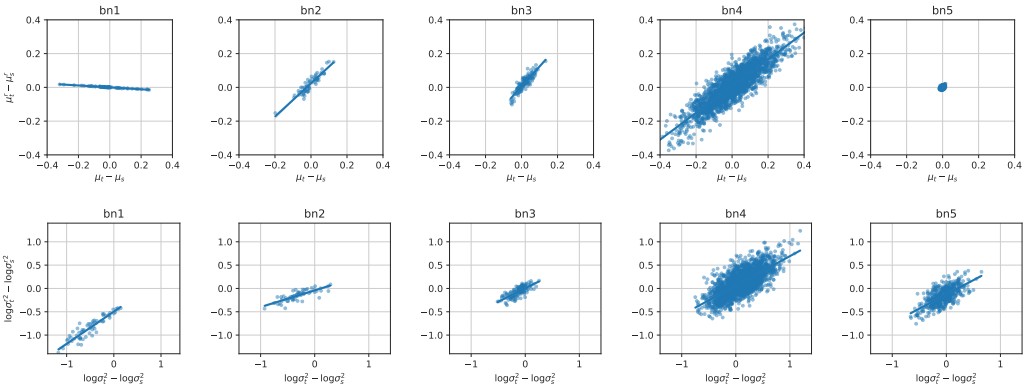

Figure 14: Correlation of statistic differences of mean (top) and log-variance (bottom) in BN layers.

## D.4 EXPERIMENTS WITH RESNET

Like Table 2, we conduct the same Domain-Net experiments but using ResNet18 (He et al., 2016) in place of AlexNet. Most configurations are the same but we use a cosine-annealing schedule of the learning rate from 0.05 to 0 within 600 epochs. Compared to

Table 15: Robustness propagation using ResNet18.

| AT users | All | | 20% | | Real | |
|---|---|---|---|---|---|---|
| Metrics | RA | SA | RA | SA | RA | SA |
| FedRBN (ours) | **59.1** | **61.6** | **49.6** | 63.1 | **42.9** | 56.4 |
| FATBN | 37.3 | 60.4 | 25.5 | 62.8 | 13.6 | **57.4** |
| FATAvg | 46.6 | 57.1 | 36.5 | 61.6 | 18.8 | 49.0 |

AlexNet, ResNet18 is significantly more robust in all three tasks. Consistent with the AlexNet-based results, our method outperforms the two best baselines.

## D.5 WHEN DOES DEBIASING WORK?

On deriving Eq. (4), we made an assumption of the independence of adversarial noise. By investigating how the assumption holds up in real datasets, we want to know when the debiasing strategy could improve the robust accuracy. For this purpose, we leverage Pearson coefficients ($\mathbb{P}$) to estimate how dependent the noise is on the input images. For each user in the Digits dataset, we calculate $R(\mu^r - \mu) = \frac{1}{L} \sum_{l=1}^{L} (\mathbb{P}(\mu_l^r - \mu_l, \mu_l) + \mathbb{P}(\mu_l^r - \mu_r, \mu_r))/2$, where $\mu_l$, $\mu_l^r$ are the statistics of the layer $l \in \{1, \ldots, L\}$. The experiment protocol follows the 100% 1/5 domain setting in Table 1.

As reported in Fig. 15, we find that the dependence weakens by layer and the RA improvement brought by debiasing is aug-

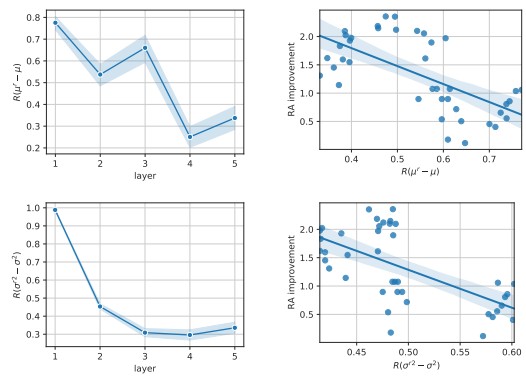

Figure 15: Evaluation of the noise-independence assumption (*left two*) and its effects on users' RA improvement from debiasing (*right two*). Estimated by the Pearson coefficient, $R$ presents the degree to which noise statistics (e.g., $\mu^r - \mu$) are correlated to data statistics (e.g., $\mu$).

mented by lower dependence, i.e., $R$. The phenomenon is related to the nature of adversarial noise. Since adversarial noise is inherently so subtle to be decoupled from noised images, the linear modeling of batch-normalization in shallow layers is less likely to extract the noise (for example, by $\mu_l^r - \mu_l$). In deeper layers, the noised and clean features deviates from each other, for example, in the logit layers (see Fig. 11). Therefore, the noise correlation is weaker for the noise-independence assumption to hold better and our debiasing method is more effective in improving propagated robustness.

