# OpenReview forum: "Federated Robustness Propagation: Sharing Adversarial Robustness in Federated Learning"
_ICLR.cc/2022/Conference — ICLR 2022 Submitted_

### Official Review · Reviewer_udUC · 2021-10-31

**Correctness:** 2
**Technical Novelty And Significance:** 3
**Empirical Novelty And Significance:** 3
**Recommendation:** 3
**Confidence:** 4

**Main Review:**

Strengths:
(1) A novel FedRBN method was proposed for adversarial robustness propagation among users in the federated learning setting.
(2) The dual batch-normalization (DBN) structure enabled communication efficiency and robustness propagation.
(3) Experiments showed that it improved the adversarial robustness of the federated learning algorithms.

Weaknesses:
(1) One major concern is the rationality of the dual batch-normalization (DBN) structure in the proposed method. The results in Table 1
indicated that “simply adding DBN does not help unless the noise detector is applied”. On the other hand, different from prior work, it
used shared weight and bias in the design of DBN. It is not clear whether the parameter sharing strategy explained the limitation of
DBN-only module in the proposed method. In addition, the difference between “+copy” and “+debias” is not clear.
(2) Compared to FATBN, FedRBN used the robustness propagation from AT user to ST user. In this case, it is not shown whether FATBN allows adversarial training for every user. If so, why is it performing worse than FedRBN with robustness propagation?
In other others, can the robustness propagation on one ST user achieve better performance than performing adversarial training on this user? Or the performance improvement can be explained by the DBN and the noise detector in FedRBN?
(3) The strategy of learning user-specific BN parameters is close to a federated multi-task learning problem [ref 1]. It might be
convincing to have some discussion on comparing this work to the federated multi-task learning solutions.
[ref 1] Smith, Virginia, Chao-Kai Chiang, Maziar Sanjabi, and Ameet Talwalkar. "Federated multi-task learning." In Proceedings of the 31st International Conference on Neural Information Processing Systems, pp. 4427-4437. 2017.
(4) It is not convincing why ST users would average the statistic difference received from all AT users. On one hand, this averaging method in FL has shown to be sensitive to Byzantine attacks.  On the other hand, it might be dominant in the major data distribution
D_i shared by most AT users in real-world scenarios. Sending the averaged statistic difference to all ST users with different
distributions could lead to a sub-optimal solution.
(5) As one of the key component of FedRBN, the training of the noise detector needs to be clarified, especially for ST users with limited
data and computational budgets. Can the noise detector be propagated from AT users to ST users in this case?
(6) The legend of Figure 3(b) is confusing.
Minor issues:
(1) Some notations are undefined, e.g., q_a and q_k.
(2) There are some grammar errors, e.g., “the corresponding two distribution, separately.”, “the server aggregate BN statistic”, etc. 

**Summary Of The Paper:**

This paper studied the propagation of adversarial robustness among federated learning users, where some users had limited training data and computational budgets for affording the adversarial training. It proposed a novel method Federated Robust Batch-Normalization (FedRBN) to facilitate this propagation. The key step was a dual batch-normalization (DBN) structure to encode the clean and adversarial distributions respectively. Then the debiased BN statistic for low-resource users was estimated from other users who could afford the expansive adversarial training. The experiments supported that this approach improved the adversarial robustness of federated learning. 

**Summary Of The Review:**

The rationality of technical details used in this paper is not well explained, thus I recommend the rejection of this work for ICLR.

---

> ### Author Response · Authors · 2021-11-20
> **Response to Reviewer udUC**
>
> We appreciate the reviewer's detailed comments. We have tried to address your questions carefully and hope you can actively participate in the discussion.
>
> > Weaknesses:
> >
> > (1) One major concern is the rationality of the dual batch-normalization (DBN) structure in the proposed method.
>
> Table 1 shows the limitation of the DBN when the adversarial BN is only locally estimated.
>
> The role of the DBN is two-fold.
> 1. With DBN, we can characterize the domain shifts by the clean BN statistics from different domains while keeping a separate robust BN.
> 2. As shown by (Xie et al., 2020), DBN speeds up the convergence of adversarial training and augments the final robustness.
> Without DBN, our method degrades as FATBN, in which case we cannot debias adaptively.
> The main reason for the failure of DBN in Table 1 is that the adversarial component of DBN does not gain proper statistics since the local computation-budget-insufficient users cannot generate adversarial samples.
>
> In our experiments, we do not observe any improvement when the affines are not shared.
>
> > In addition, the difference between “+copy” and “+debias” is not clear.
>
> `+copy` means copying BN statistics without  `debias`ing the copied BN towards the local domain (Eq. (4)).
>
> > (2) Compared to FATBN, FedRBN used the robustness propagation from AT user to ST user. In this case, it is not shown whether FATBN allows adversarial training for every user. If so, why is it performing worse than FedRBN with robustness propagation? In other others, can the robustness propagation on one ST user achieve better performance than performing adversarial training on this user? Or the performance improvement can be explained by the DBN and the noise detector in FedRBN?
>
>
> For a fair comparison, we execute all algorithms under an identical setting.
> For example, we let "20%-per-domain users from 3/5 domains (of Digits) conduct AT" (quote from the last sentence of the first paragraph in Sec. 5.1).
>
> The robustness propagation (i.e., partial users do AT) cannot perform as well as the full adversarial training (i.e., all users do AT).
> One evidence can be found in Fig 1b where the propagated RA (all results with less than 100% portion of AT users) is lower than fully AT (100% AT users).
> The results demonstrate the difficulty of the proposed novel problem setting (Federated Robustness Propagation).
>
> The performance improvement cannot be explained by the DBN and the noise detector in FedRBN as shown in the ablation study (Table 1 and the paragraph above Table 1).
>
> > (3) The strategy of learning user-specific BN parameters is close to a federated multi-task learning problem [ref 1].
>
> We appreciate the reviewer mentioning the connection between our work and the federated MTL.
> We have reviewed some work that adapts models for local data distributions or tasks **in the 2nd paragraph of Sec. 1 and the middle of the 1st paragraph**.
> We argue that our work is quite distinguishable from [ref 1].
> [ref 1] only studied the linear cases with regularization on the parameters, while we studied more general deep neural networks.
> In addition, [ref 1] did not consider an adversarial loss or regularization.
> Their regularization is independent of the data but depends on the task relations.
> We add the discussion in the last paragraph of Appendix A.
>
> > (4) It is not convincing why ST users would average the statistic difference received from all AT users.
>
> Thank you for your suggestions.
> On page 14, we show that a re-weighting strategy could be helpful for mitigating the distribution difference of source domains.
>
> > (5) As one of the key component of FedRBN, the training of the noise detector needs to be clarified, especially for ST users with limited data and computational budgets. Can the noise detector be propagated from AT users to ST users in this case?
>
> As stated in Alg. 3, the adversarial samples are generated locally, the logits of adversarial and clean samples are computed, and then a noise detector is fitted.
> For ST users, we conduct the same training strategy as Alg. 3.
> This may indeed increase the computation budget for ST users, for example, additional time and memory consumption.
> But we argue that such an increase is marginal.
> In Table 2, we report the training time (including the time of noise detector) in the column $T$, where we see that
> the difference between FedRBN and baselines (e.g., FedBN) is very small.
> For example, 665 of FedRBN versus 663 of FedBN.
>
> We thank the reviewer for the suggestion of propagating the noise detector.
> We believe a propagated noise detector may *further*  improve the efficiency of FedRBN.
> But we want to remark that the possible improvement will not diminish the claims and contributions of the paper, for example, FedRBN is the desired solution for propagating robustness.

---

> ### Author Response · Authors · 2021-11-27
> **A kind reminder**
>
> Dear Reviewer udUC,
>
> Thank you for your time to read our paper and your valuable suggestions are important for improving the quality of our paper. As the discussion end time is close, we want to make sure all your concerns are addressed on time if any.
>
> Authors

---

### Official Review · Reviewer_B9Be · 2021-11-01

**Correctness:** 3
**Technical Novelty And Significance:** 3
**Empirical Novelty And Significance:** 2
**Recommendation:** 6
**Confidence:** 3

**Main Review:**

# Strong Points
* As for the centralized case, the robustness issue is also important in the federated learning context. Hence, this paper addresses an important concern.
* The concept of some parties having more resources than others and can therefore afford to perform adversarial training is reasonable and realistic.
* The paper is overall well-written and easy to follow.
* The authors provide extensive experimental results.

# Weak Points
* The theoretical novelty of this work is somewhat limited. In essence, the authors just apply previous insights, such as the distributional shift caused by BN and adversarial training to the context of federated learning. The authors did not reveal any new insights regarding these topics which are specific to the federated learning topic.
* The proposed approach uses a dual batch-normalization (DBN) strategy. To decide which one of the BN statistics to use, the authors state that they “predict whether the image is noised”. Here, I identify a fundamental flaw of this work, since previous works showed, that most adversarial detection techniques can be fooled [1,2,3,4], e.g. due to obfuscated gradients. Different guidelines had been proposed to check against these techniques [2,3]. The proposed noise detector seems to be a DNN-based binary classifier, similar to the approach in [5]. Hence, a simple idea for a full white-box evaluation would be to include a loss term with the objective to fool the noise detector in the adversarial example generation. This issue is critical since the noise detector appears to be crucial for the efficacy of the DBN as seen from Table 1 and stated by the authors: “DBN does not help unless the noise detector is applied”.
* The authors state that “The black-box can avoid the trip of fake robustness due to obfuscated gradient”. I do not agree with this statement, since usually, a black-box attack is harder than a white-box attack. A good start to test against obfuscated gradients would be to evaluate with Backward Pass Differentiable Approximation (BPDA) [2].
* The used baselines are not clear to me. The authors state that they “use three representative federated baselines combined with AT”, but also “denote the AT-augmented” version with a FAT prefix. From this description, the difference between FedAvg and FATAvg is not clear. Due to this confusion, I assumed that the methods with the Fed prefix refer to the paper baselines, while the AT variants are denoted with the FAT prefix. This might be my fault for not understanding something here, but I would be happy if the authors could clarify this in the rebuttal, such that I can clearly understand the results.
It would also be good to add citations to the corresponding technique in the Tables for clarity, or if not the exact method is used to clearly state the differences to the previous technique and the reasons for the introduced changes.
From Table 2 it appears that the standard techniques are already relatively robust against adversarial attacks. Commonly I would assume that a “non-protected” method (FedBN, FedAvg, FedProx) would have a RA of around 0%. Why is this?
Further, Table 2 compares two other “robust” methods (FATMeta & FedRob). These methods achieve the worst results. This result also seems counterintuitive and requires a detailed discussion in my opinion.
The same applies in Table 3. Why do FATAvg and FATBN show lower robust performance than FedAvg and FedBN? Why do the numbers of FedRBN from Table 2 and Table 3 not match?
I am also missing a comparison with FAT (Zizzo et al., 2020;)
* The choice of AlexNet for the DomainNet experiments is outdated, especially since BN has to be added. I recommend the authors to include results at least for a ResNet architecture.

[1] Adversarial Examples Are Not Easily Detected: Bypassing Ten Detection Methods; AISec’17
[2] Obfuscated Gradients Give a False Sense of Security: Circumventing Defenses to Adversarial Examples; ICML 2018
[3] On Evaluating Adversarial Robustness; arXiv 2019
[4] Detecting Adversarial Examples Is (Nearly) As Hard As Classifying Them; ICML 2021 workshop
[5] On Detecting Adversarial Perturbations; ICLR 2017


# Additional Questions
* Will the code be publicly available?


**Summary Of The Paper:**

This work studies adversarial robustness in a federated learning context. The authors propose to propagate robustness through sharing BN statistics of the different federated learning participants. Hereby, the authors consider that some parties are stronger (e.g. through higher resource power) to perform adversarial training, and can provide weaker parties with the respective parameter updates.

**Summary Of The Review:**

Robustness is an important issue also in the federated learning context. Overall I identify the noise detector as the main bottleneck of this work since a “smart” adversary can incorporate it into its attack strategy and can therefore probably significantly deteriorate the entire FedRBN pipeline. Also, the main results appear confusing to me and require some further discussion.

======= Edit November 23 =========
Raised my score to "marginally above the acceptance threshold" after reading the rebuttal.

---

> ### Author Response · Authors · 2021-11-20
> **Response to Reviewer B9Be on the novelty**
>
> We thank the reviewer for acknowledging the importance of our work. We would like to re-iterate our novelty here and clarify some misunderstood points later.
>
> > Weak Points
> >
> > * The theoretical novelty of this work is somewhat limited. In essence, the authors just apply previous insights, such as the distributional shift caused by BN and adversarial training to the context of federated learning. The authors did not reveal any new insights regarding these topics which are specific to the federated learning topic.
>
> We agree with the reviewer on the fact that our work is inspired by prior work: the FedBN for using local batch-normalizations (BNs) and dual batch-normalization (DBN) for decoupling adversarial and clean samples, as we have stated in Section 4.1 (before Eq. (3)).
> However, our work is the first to formally propose and solve the practical “Federated Robustness Propagation” problem. Our proposed solution is technically novel instead of a naive combination of previous methods.
> Below, we want to reiterate the two aspects of our contributions (which have already been discussed **in the last two paragraphs of Page 2**).
>
> **(1)** Novelty in problem setting: we illustrated a novel yet practical problem setting and reveal its challenges: Federated Robustness Propagation **(the last 2nd paragraph of Sec. 1 and Fig 1a,b)**.
> The varying capability for computation-intensive robust training ubiquitously exists in federated users due to diverse hardware, but how model robustness will be degraded has not been studied yet.
> Remarkably, trained by traditional federated methods, the robust accuracy drops by over 15% (or 25% relative to the full-trained RA) if only 20% of users can afford robust training.
> **The conflict between the energy-efficient hardware and the desires for on-device adversarial robustness has not received attention until this work** and calls for better robustness-propagation methods.
>
> **(2)** Novelty in technical solutions: we presented the first federated algorithm to address the low efficiency of robustness propagation from the AT users to ST users **(see Fig 1b and the last paragraph of Sec. 1b)**, which is different from both previous FL and AT methods.
>
> DBN aims to **improve** model robustness by disentangling clean and adversarial BN statistics in adversarial training. In contrast, we aim to **propagate** adversarial robustness learned on resource-sufficient one node to a resource-insufficient one. The most significant technical difference is we have *asymmetric learning*: some nodes do standard training while others do adversarial training. Both previous centralized AT and federated AT papers consider a *symmetric learning* system: all nodes are doing AT. Such an asymmetric learning system is motivated by practical requirements (see the first bullet). To propagate robustness in such an asymmetric system, we designed a novel robustness propagation rule: *the affine transformation rule*.
> An intuitive explanation of our method is provided based on an affine-noise assumption in the last paragraph of Sec. 4.1.
> Worth noticing, we empirically demonstrated that the approximated debiasing together with DBN copying propagates robustness efficiently across domains: **only less than 15% of the full-trained RA is lost after propagation from 20% AT-affordable users.**

---

> ### Author Response · Authors · 2021-11-20
> **Response to Reviewer B9Be on the major concerns**
>
> We would like to address the major concerns in this reply.
>
> > * ... that most adversarial detection techniques can be fooled [1,2,3,4], e.g. due to obfuscated gradients. ... evaluate by including a loss term with the objective to fool the noise detector in the adversarial example generation.
>
> We agree with the reviewer that the noise detector can be fooled, which has already been discussed in Appendix C.8 (of our first submission).
> As we mentioned in **the end of Paragraph 2 of Page 8**, the joint attack, *jointly maximizing classification loss and noise detection loss*, will degrade the detection accuracy somehow but will not weaken the advantage of our method against baselines.
> Though the detection is degraded (see **Table 13**) by the joint attack (for example, 24% accuracy drop for 20% AT Digits setting), the accuracy of our method is still outstanding (see **Table 12**). For example, 50.6% versus the best baseline 42.6% in the 20% AT Digits setting.
> Other attacks are evaluated in **Table 4**, including the currently strongest auto-attack (AA) and black-box attack (LSA).
> All these evidences have supported the effectiveness of our method with both DBN and noise detector.
>
> > * The authors state that “The black-box can avoid the trip of fake robustness due to obfuscated gradient”. I do not agree with this statement, since usually, a black-box attack is harder than a white-box attack. ... evaluate with BPDA [2].
>
> In our revision, we updated the statement to a more solid one: “strong score-based blackbox attacks such as Square Attack can avoid the trip fake robustness due to obfuscated gradient (Andriushchenko et al, 2019)”.
>
>
> We are aware of BPDA but did not use it in our experiments since it is not applicable. BPDA is used to avoid the gradient masking effect caused by random or non-differentiable operations in defense algorithms [2].
>
> We tried to avoid fake robustness by using adaptive attack and AutoAttack:
>
> Our adaptive attack, which jointly attacks the adversarial detector and classifier, is an effective attack method to avoid fake robustness: "With the concern that the attacker may bypass the noise detector and lead to the trip of fake robustness (Athalye et al. 2018), we include a joint-attack on the noise detector and the model prediction in Appendix C.8."
>
> We also evaluated the model robustness using Auto-Attack (AA), which is the strongest adversarial attack and has been shown to be effective in avoiding fake gradients caused by gradient masking effects (Croce&Hein, 2020).
>
> > * AlexNet is outdated for DomainNet
>
> We followed previous work (Li et al., 2020c) to design the federated experiments.
> In our revision, we also add the results with ResNet18 in Table 15 and Appendix. D.4.
> Compared to AlexNet, ResNet18 is significantly more robust in all three tasks.
> Consistent with the AlexNet-based results, our method outperformed the two baselines.
>
> | Method | RA (All) | SA (All) | RA (20%) | SA (20%) | RA (Real) | SA (Real) |
> | :----- | :----- | :----- | :----- | :----- | :----- | :----- |
> | FedRBN (ours) |  **59.1** | 61.6 | **49.6** | **63.1** | **42.9** | 56.4   |
> | FATBN         |  37.3 | 60.4 | 25.5 | 62.8 | 13.6 | **57.4**   |
> | FATAvg        |  46.6 | 57.1 | 36.5 | 61.6 | 18.8 | 49.0 |
>
>
> > Will the code be publicly available?
>
> Yes. We will release codes upon acceptance.

---

> > ### Comment · Reviewer_B9Be · 2021-11-23
> > **Thanks for the response**
> >
> > I thank the authors for addressing all my concerns. I am happy to see the clarifications and hope they will all be properly reflected in the final version. Please keep the readers in mind and make everything as simple to understand as possible.
> > Since most of my concerns were addressed I am raising my score to "above the acceptance threshold". I cannot give an "accept" due to my concerns regarding the novelty. However, I believe that this work can contribute to the federated learning community.
> > Best regards
> > Reviewer B9Be

---

> > > ### Author Response · Authors · 2021-11-23
> > > **Response to Reviewer B9Be**
> > >
> > > Dear Reviewer B9Be,
> > >
> > > Many thanks for all the helpful comments and positive re-assessment. We really appreciate you for increasing our score. We totally agree keeping things simple is important and we will try our best to revise the paper in the final version accordingly.
> > >
> > > Best wishes,
> > >
> > > Authors

---

> ### Author Response · Authors · 2021-11-20
> **Response to Reviewer B9Be with clarifications in experiment**
>
> We appreciate the reviewers' detailed comments. We want to clarify some points that may be misleading in our experiments.
>
> > * The names of used baselines are not clear to me. ... add citations to the corresponding technique in the Tables for clarity
>
> We appreciate the reviewer raising the concern and we update the table accordingly.
> In the new revision, we update Table 2 with FAT-prefixes which indicate the extensions of  base methods with adversarial training (based on PGD7 attack).
> A special case is FedRob which does not use PGD attack for adversarial training but an affine noise in a similar adversarial manner.
> In the initial version, we used the FedAvg, FedProx such that readers can easily recap their original work when many methods are listed.
> We do not include the citations in Table 2 because the full-name citation is too long to be included in such a small space.
>
>
> > * From Table 2 it appears that the standard techniques are already relatively robust against adversarial attacks. Commonly I would assume that a “non-protected” method (FedBN, FedAvg, FedProx) would have a RA of around 0%. Why is this?
>
> We believe the comment results from the possible confusion of the method names.
> Actually, there are no unprotected methods in Table 2 but only protected ones (FedBN->FATBN, FedAvg->FATAvg, FedProx->FATProx).
> We hope such clarification resolves the confusion.
>
> > * Further, Table 2 compares two other “robust” methods (FATMeta & FedRob). These methods achieve the worst results. This result also seems counterintuitive and requires a detailed discussion in my opinion.
>
> The two methods involved complicated update strategies.
> Especially, they require second-order information, e.g., Hessian or one-step-ahead gradient.
> Therefore, convergence is difficult for tasks with significant distribution shifts (different domains) and different objectives (AT versus ST).
>
> > * The same applies in Table 3. Why do FATAvg and FATBN show lower robust performance than FedAvg and FedBN? Why do the numbers of FedRBN from Table 2 and Table 3 not match?
>
> Table 2 and Table 3 have different settings.
> In Table 3, we used 20% users from 3 domains only (20% 3/5 AT domains) for AT, while in Table 2, we either use 20% users from 5 domains (denoted as 20%) or all users from a single domain (denoted as MNIST/Real).
> The `20% 3/5 AT domains` setting not only induces cross-domain propagation but also in-domain propagation.
> Therefore, the differences in settings may lead to the differences in the reported RA or SA.
>
> > * missing FAT (Zizzo et al., 2020;)
>
> Since FAT is a combination of FedAvg and AT, in our paper it is named FATAvg.

---

### Official Review · Reviewer_bKE4 · 2021-11-01

**Correctness:** 4
**Technical Novelty And Significance:** 3
**Empirical Novelty And Significance:** 3
**Recommendation:** 8
**Confidence:** 4

**Main Review:**

This paper aims to improve the adversarial robustness of federated learning with heterogeneous users. The authors consider both the statistical heterogeneity and the computational heterogeneity in federated learning, which is a realistic scenario. Considering the heterogeneous computational resource on different users, they propose to use a small fraction of clients for adversarial training, and copy the BN layers of these clients to others who could not afford for adversarial training to propagate the adversarial robustness. The method is simple and the experiments show that it is effective.

Strengths:
1. The paper studies an important trustworthy AI topic: preserving both privacy and robustness via a new federated robust training scheme. Since robust training is often too expensive for edge users, the authors studied a novel setting of practical relevance: users with limited restricted budgets only afford cheaper standard local training and obtain robustness from other budget-sufficient users who use adversarial training.

2. The proposed method is simple and intuitive. They take advantage of the idea from the previous work that the robustness is highly correlated with the BN layers, and propose to propagate the robustness from a robust user to a non-robust user with linear combination of the BN statistic. The authors also setup a verification experiment to show that the linear combination is reasonable.

3. The evaluation is very thorough and convincing. Detailed experiments involving multitude of datasets from different domains (to capture data heterogeneity). To fully evaluate the robustness, the authors experiment with (black-box) MIA, (adaptive) AutoAttack and LSA. The method still outperforms the baselines by a large margin under adaptive attacks. I also appreciate authors providing standard deviations for results.

Weaknesses:
1. In the last paragraph of Sec 4.1, the authors assumes the noise in each layer is independent of the data: I don’t understand this assumption since adversarial noise is data-dependent.

2. Since the BN layers are only updated locally with limited data and/or SGD rounds, it is unclear to me why the BN parameters can be sufficient to capture the robustness property?

3. The proposed method mainly focuses on the feature heterogeneity. I would like to know whether FedRBN still work under label heterogeneity, which could be a more challenging heterogeneous scenario.

4. Have the authors tried to scale to more clients, or more skewed local data distributions?

**Summary Of The Paper:**

This paper proposes a simple yet efficient method to improve the adversarial robustness of federated learning. The proposed method allows only a subset of clients to perform computationally expensive adversarial training, and then propagate the robustness on these clients to the others who could not afford the adversarial training. Extensive experiments are conducted and the results show that the proposed method outperforms other baselines.

**Summary Of The Review:**

This work has strong motivation and adds novel contributions to federated learning, i.e. studying federated learning robustness under heterogeneity, The proposed model is simple yet easy to understand, and is theoretically grounded. Experiments are convincing, although a few questions remain to be clarified by authors.

---

> ### Author Response · Authors · 2021-11-20
> **Response to Reviewer bKE4**
>
> Thanks for acknowledging the importance of the problem we studied and the novelty of our new fairness metric. Below we provide detailed answers to your questions. We hope they address your concerns and make you more convinced about the contributions of our work.
>
> > Weaknesses:
> >
> > In the last paragraph of Sec 4.1, the authors assume the noise in each layer is independent of the data: I don’t understand this assumption since adversarial noise is data-dependent.
>
> First, we want to remark that the assumption serves the sole purpose of an intuition explanation.
> Second, we empirically observe the correlation which is approximately linear as we expected.
> The observation empirically supports the assumption for Eq (4).
> We would like to look for more rigorous proof in the future.
>
> > Since the BN layers are only updated locally with limited data and/or SGD rounds, it is unclear to me why the BN parameters can be sufficient to capture the robustness property?
>
> We had a similar concern about the scarcity of local data. Therefore, we have conducted experiments with varying sample sizes in Fig 9 of Appendix C.3 (from our initial version).
> As demonstrated in Fig. 9, our method empirically outperforms the baselines even when only 10% of the Digits dataset is available.
> Note that with 10% of Digits dataset, each user will only have 74 samples, a relatively small sample size.
>
> One important reason for the success of BN in modeling adversarial distributions is that the BN statistics are estimated by exponential averaging of iterations.
> Since the adversarial samples are crafted at every batch gradient descent, the exponential averaging will always take new features into account.
> Therefore, even with the limited samples, the performance could be relatively sweet.
>
>
> > The proposed method mainly focuses on feature heterogeneity. I would like to know whether FedRBN still works under label heterogeneity, which could be a more challenging heterogeneous scenario.
>
> We follow previous work (Li et al., 2020c) to conduct the non-iid experiments. We would prefer to discuss the label heterogeneity problem in the future.
>
> > Have the authors tried to scale to more clients, or more skewed local data distributions?
>
> To evaluate the performance of our method with more users, we add experiments in *Appendix D.2* where we scale the number of clients from 25 (smaller than our benchmarks) to 200 (much larger than our benchmarks).
> Moreover, we randomly choose different numbers of samples for each user in a non-iid manner.
> In the worst case, some clients will only have 32 samples for training.
> Given the same sample size in total, distributing data to *more clients results in worse performance* either in the sense of robustness or accuracy (see Fig. 13).
> As observed in the convergence curves, when data are more concentrated in a few users (i.e. smaller numbers of users), the convergence will be faster, and therefore fewer communication rounds are required.

---

### Official Review · Reviewer_dgjR · 2021-11-03

**Correctness:** 3
**Technical Novelty And Significance:** 3
**Empirical Novelty And Significance:** 3
**Recommendation:** 8
**Confidence:** 4

**Main Review:**

Making model robust against adversaries is very important for deploying FL. The authors present a practical and effective approach to tackle this issue. It is interesting to propagate adversarial robustness via transfer of batch-normalization statistics from high-resource users to low-resource users. The effectiveness of the proposed method is validated by extensive experiments and ablation studies. The paper is well-written and easy to follow overall.

The idea of debiasing is not very clear. Eq 4 can work well if the noise values at both users follow the same distribution as described in page 5. However, can Eq. 4 really help when the noise distributions are not the same? The effect of debiasing seems to be marginal in
Table 1. I want to see more experimental results on this particular issue (ablation on when debaising works well versus when does not), as it is a key component of the proposed idea.

While writing is good overall, certain points are vague. Explanations and motivations related to Fig. 2 should be made more clear. For example, Fig. 2a is presented to argue there are significant differences in BN parameters between ST and AT users from the same domain. But this is incomprehensible. Also, is there any reason why BN2 is picked for Fig. 2b? Are the correlations similar across BN layers?

There seem to be significant enough departure from a simple linear relationship between the delta log variance of clean samples vs that of perturbed samples in Fig. 2b. Yet the algorithm gives good performance overall. I would like to see some sort of convergence analysis that can give insights into, e.g., the effect of uncertainly in such linear modeling on the convergence behavior. Note, most FL paper today provide analysis to ensure a controllable convergence behavior. The authors can build on existing convergence analysis.

Among 50 users in the system, how many users participate in FL at each round? This is not clearly described, and I believe the authors assume all users participate in each round. How does the proposed method perform when the contact ratio is small (e.g., 20 or 30 users out of 100, per round), which reflects the practical cross-device federated learning setup? The global statistics on batch-normalization can also be biased when the contact ratio is less than 1.

Also, the total number of devices is set to 30 or 50 depending on datasets in the presented experiments. But standard FL tasks require large-scale learning. Can you comment on the scalability of your method? (specifically, would it perform well in the presence of more devices?)





**Summary Of The Paper:**

Adversarial training (AT) is good for defending against attacks but is costly for low-resource FL users. The authors want to find a good way to propagate adversarial robustness from high-resource users to low-resource ones. They proposes federated robust batch-normalization (FedRBN) which enables all users to enjoy robustness against malicious attacks at inference. The basic idea is to let the BN statistics of AT users be shared with the standard users so that the latter can do robust prediction on adversarial inputs without conducting AT on their own. The authors also propose a debiasing method to mitigate the effect of non-IID data distributions across the clients. Experimental results including various ablation studies confirm the advantage of FedRBN.


**Summary Of The Review:**

Overall a good paper, but I am still giving the “marginally above” score due to some questions/concerns as given above; however, upon getting reasonably satisfactory answers, especially to the convergence analysis question, I will be happy to move higher to the positive side.

---

> ### Author Response · Authors · 2021-11-20
> **Response to Reviewer dgjR**
>
> Thanks for acknowledging the importance of the problem we studied and the novelty of our new fairness metric. Below we provide detailed answers to your questions.
>
> > ... can Eq. 4 really help when the noise distributions are not the same? The effect of debiasing seems to be marginal in Table 1. I want to see more experimental results on this particular issue (ablation on when debaising works well versus when does not)
>
> In Fig. 15 of Appendix D.5, we investigate the relation between the noise-dependence and the robustness improvement brought by debiasing.
> When per-layer adversarial noise is less dependent on the features, the debiasing tends to improve robustness more.
> The observation is consistent with our analysis for Eq. (4) where we assume the noise are independent from features.
>
>
> > While writing is good overall, certain points are vague. Explanations and motivations related to Fig. 2 should be made more clear. For example, Fig. 2a is presented to argue there are significant differences in BN parameters between ST and AT users from the same domain. But this is incomprehensible. Also, is there any reason why BN2 is picked for Fig. 2b? Are the correlations similar across BN layers?
>
> In Fig 2a, we show the differences are non-zero in the y-axis. To show the differences are significant, we plot the difference against the clean statistics. We update the description to clarify the point.
>
> We present Fig 2b because it is a strong example demonstrating our intuition.
> To further illustrate the differences, we have provided more correlation plots in  Appendix D.3.
> In most layers, the linear correlation is significant. Though the last layer presents more randomness, the significance of the values is weak.
>
> > There seem to be significant enough departure from a simple linear relationship between the delta log variance of clean samples vs that of perturbed samples in Fig. 2b. Yet the algorithm gives good performance overall. I would like to see some sort of convergence analysis that can give insights into, e.g., the effect of uncertainly in such linear modeling on the convergence behavior. Note, most FL paper today provide analysis to ensure a controllable convergence behavior. The authors can build on existing convergence analysis.
>
> We want to clarify that the BN copying and debiasing (Alg. 2) is only **executed after federated training**.
> Though the motivation experiments (i.e., the linear modeling in Fig. 2b) present large uncertainty, it does **not affect** the convergence.
> Since our new method does not introduce more uncertainty in the convergence than the baselines, the convergence should be similar to FedBN which has been theoretically proved.
>
> > Among 50 users in the system, how many users participate in FL at each round? This is not clearly described, and I believe the authors assume all users participate in each round. How does the proposed method perform when the contact ratio is small ..., which reflects the practical cross-device federated learning setup? The global statistics on batch-normalization can also be biased when the contact ratio is less than 1.
>
> Following (Li et al. 2020c), we assume that all users participate in FL per round, which has been clarified in our detailed experiment descriptions in Appendix C.1 (after revision).
> In addition, we added experiments to simulate the scenario with low contact ratios (from 2 to 40 out of 50 users) in Appendix D.1.
> Interestingly, our results showed that a lower contact ratio leads to higher robustness when standard accuracy is maintained similarly.
> For example, the robust accuracy is 59% with a contact ratio of 2/50 but 56% with a contact ratio of 2/50.
> The phenomenon is observed both for all methods (ours, FATAvg, and FATBN).
> Since the actual update steps in the view of the global server are reduced with lower contact ratios, the result is consistent with Table 5&14, where smaller batch sizes or fewer local steps lead to better robustness.
>
> > Also, the total number of devices is set to 30 or 50 depending on datasets in the presented experiments. But standard FL tasks require large-scale learning. Can you comment on the scalability of your method? (specifically, would it perform well in the presence of more devices?)
>
> Since our method has the similar training/communication strategy as FATBN or FATAvg (except switching BN), the federation can scale up to more clients like FATBN or FATAvg who are widely used scalable implementations.
> To evaluate the performance of our method with more users, we add experiments in Appendix D.2 where we scale the number of clients from 25 (smaller than our benchmarks) to 200 (much larger than our benchmarks).
> Distributing data to *more clients results in worse performance* either in the sense of robustness or accuracy.
> As observed in the convergence curves, when data are more concentrated in a few users (i.e. smaller numbers of users), the convergence will be faster, and therefore fewer communication rounds are required.

---

> ### Comment · Reviewer_dgjR · 2021-11-30
> **review update:**
>
> Most questions (albeit not all) have been addressed adequately. I raise my score to "Accept".

---

### Official Review · Reviewer_p9LS · 2021-11-07

**Correctness:** 4
**Technical Novelty And Significance:** 2
**Empirical Novelty And Significance:** 2
**Recommendation:** 3
**Confidence:** 4

**Main Review:**

Pros:
a. The authors propose Federated Batch-Normalization (FedRBN) to transfer robustness between users participating in FL. They build on a prior observation (in the centralized setting) that robustness is correlated with BN statistics and hence copying BN parameters could lead to transferring robustness.

b. The authors decouple the effect on BN parameters to (1) Domain difference (due to non-iid data) and (2) Adversarial noise (due to AT). To counter this they propose to use a dual batch-normalization structure, which keeps track of two sets of BN statistics: one for clean data and another for noisy data (due to AT).

c. Finally, they propose a de-biased estimation of the robust BN statistics (mean, variance) for the ST users (Eq. (4)) which leads to the transfer of robustness from users in AT to users in ST. The authors use a one-dimensional example to justify their choice of expression in (4).

d. I like the choice of datasets used by the authors to perform the evaluation.

Cons:
1.  The paper needs a careful editorial pass. There are numerous typos, and certain parts of the paper could benefit immensely from more clarity in writing.

2. The contributions of this paper are marginal. For example, prior work (FedBN) has shown that the federation of BN parameters leads to poor model accuracy. Hence, these parameters are only locally optimized. Similarly, the connection between BN parameters and robustness has also been made. This paper proposes a mechanism to transfer BN statistics between users. Furthermore, the linear mapping is not rigorously justified.

3. The post-training modification of the BN statistics and its effect on model performance seems to be heavily dependent on the model-wise switch parameter $h$ (defined on pg 5). In this context, Algorithm 3 (in the appendix) is very important. If it is not able to separate a being sample from the adversarial one then the performance would be impacted. However, confident detection of adversarial examples is still an area of active research. I would really like to see more evidence of the performance of Algorithm 3 for stronger adversarial attacks.

4. Similarly, AA has been shown to be a stronger form of attack than PGD. It is not clear to me why a much weaker attack (PGD, n=7) was used for training the models. In fact, I do not completely understand how Table 4 has been generated.

Clarifications:
a. What is the definition of $q_k$ in Sec 3.2.
b Shouldn't (2) be better expressed using ${\bf q}$
c. In Algorithm 2, sometimes $\sigma_s^{r^2}$ is used whereas at other times $\sigma^{2^r}$ is used. It would be great if there could be more consistency in notation.
d. Some claims, for example, the authors mention that the copy of the de-biased BN statistics does not leak more information compared to FedAvg. However, no justification has been made for it. I understand that it could be orthogonal to the problem under study, but the claim seem to be a fairly important one and has not been justified.
e. Readability would be improved if the authors explicitly associate operations such as +detector, +copy, +debias (used in the ablation study) with their mathematical descriptions provided earlier in the paper.


**Summary Of The Paper:**

The paper studies the problem of propagating adversarial robustness in federated learning. Adversarial Training (AT) is typically used for training models that are robust to adversarial examples, however, AT requires computing resources that are not always available at all devices participating in FL. The paper proposes a strategy to transfer adversarial robustness among a group of FL users (characterized by non-iid data and heterogeneity in available computing resources, allowing only a fraction of them to perform AT).

**Summary Of The Review:**

Please see the detailed comments above.

---

> ### Author Response · Authors · 2021-11-20
> **Response to Reviewer p9LS on the paper contribution**
>
> We appreciate the reviewer's patient reading and comments. We have tried our best to address your concerns as below.
>
> > **Main Review**
> >
> > The contributions of this paper are marginal. Both FedBN and DBN are not new...
>
> We agree with the reviewer on the fact that our work is inspired by prior work: the FedBN for using local batch-normalizations (BNs) and dual batch-normalization (DBN) for decoupling adversarial and clean samples, as we have stated in Section 4.1 (before Eq. (3)).
> However, our work is the first to formally propose and solve the practical “Federated Robustness Propagation” problem. Our proposed solution is technically novel instead of a naive combination of previous methods.
> Below, we want to reiterate the two aspects of our contributions (which have already been discussed **in the last two paragraphs of Page 2**).
>
> **(1)** Novelty in problem setting: we illustrated a novel yet practical problem setting and reveal its challenges: Federated Robustness Propagation **(the last 2nd paragraph of Sec. 1 and Fig 1a,b)**.
> The varying capability for computation-intensive robust training ubiquitously exists in federated users due to diverse hardware, but how model robustness will be degraded has not been studied yet.
> Remarkably, trained by traditional federated methods, the robust accuracy drops by over 15% (or 25% relative to the full-trained RA) if only 20% of users can afford robust training.
> **The conflict between the energy-efficient hardware and the desires for on-device adversarial robustness has not received attention until this work** and calls for better robustness-propagation methods.
>
> **(2)** Novelty in technical solutions: we presented the first federated algorithm to address the low efficiency of robustness propagation from the AT users to ST users **(see Fig 1b and the last paragraph of Sec. 1b)**, which is different from both previous FL and AT methods.
>
> DBN aims to **improve** model robustness by disentangling clean and adversarial BN statistics in adversarial training. In contrast, we aim to **propagate** adversarial robustness learned on resource-sufficient one node to a resource-insufficient one. The most significant technical difference is we have *asymmetric learning*: some nodes do standard training while others do adversarial training. Both previous centralized AT and federated AT papers consider a *symmetric learning* system: all nodes are doing AT. Such an asymmetric learning system is motivated by practical requirements (see the first bullet). To propagate robustness in such an asymmetric system, we designed a novel robustness propagation rule: *the affine transformation rule*.
> An intuitive explanation of our method is provided based on an affine-noise assumption in the last paragraph of Sec. 4.1.
> Worth noticing, we empirically demonstrated that the approximated debiasing together with DBN copying propagates robustness efficiently across domains: **only less than 15% of the full-trained RA is lost after propagation from 20% AT-affordable users.**

---

> ### Author Response · Authors · 2021-11-20
> **Response to Reviewer p9LS on detailed comments**
>
>
> > The post-training modification of the BN statistics ... If it is not able to separate a being sample from the adversarial one then the performance would be impacted... more evidence of the performance of Algorithm 3 for stronger adversarial attacks.
>
> We have previously noted the problem and have discussed it at **the end of Paragraph 2 of Page 8**.
> In short, even if adversarial detection does not work well, our method can achieve better robust accuracy than the baselines.
> At the end of Paragraph 2 of Page 8, we have already included experimental results regarding a joint attack: Jointly maximizing the loss of task and noise detector.
> Though the detection is degraded (see **Table 13**) by the joint attack (for example, 24% accuracy drop for 20% AT Digits setting), the accuracy of our method is still outstanding (see **Table 12**), for example, 50.6% versus the best baseline 42.6% in the 20% AT Digits setting.
> Other attacks are evaluated in **Table 4**, including the currently strongest auto-attack (AA) and black-box attack (LSA).
> All these contents are included in our initial submission.
>
>
> > AA is a stronger than PGD... why a much weaker attack (PGD, n=7) was used for training the models.... how Table 4 has been generated.
>
>
> PGD has been extensively used for robust training (Zhang et al., 2019), (Madry et al., 2018) and (Chen et al., 2020) and so far whereas AA was not studied for robust training. Even though AA is stronger in terms of attack effectiveness, one possible reason for not being widely adopted in robustness learning is that AA is computational extensive and may not be efficient enough for many learning problems.
>
> * (Zhang et al., 2019) Hongyang Zhang, Yaodong Yu, Jiantao Jiao, Eric P Xing, Laurent El Ghaoui, and Michael I Jordan. Theoretically principled trade-off between robustness and accuracy. In ICML, 2019.
> * (Madry et al., 2018) Aleksander Madry, Aleksandar Makelov, Ludwig Schmidt, Dimitris Tsipras, and Adrian Vladu. Towards deep learning models resistant to adversarial attacks. ICLR, 2018.
> * (Chen et al., 2020) Tianlong Chen, Sijia Liu, Shiyu Chang, Yu Cheng, Lisa Amini, and Zhangyang Wang. Ad- versarial robustness: From self-supervised pre-training to fine-tuning. CVPR, 2020.
>
> Table 4 is generated by following previous routines in AT. Specifically, we conduct adversarial training using PGD attack (following the settings in (Madry et al., 2018)) and evaluate the adversarially trained models using multiple different adversarial attacks including PGD and AA.
> All the training strategies are consistent through our experiment section, and the AT/ST users are chosen by the benchmark-setting (see the first paragraph of Sec. 5.1): "20%-per-domain users from 3/5 domains conduct AT".
> The only variable in Table 4 is the attack method: stronger white-box, black-box methods.
> Information on the attacks is included in the second paragraph of Page 8.
>
>
> > **Clarifications**
> >
> > d. the copy of the de-biased BN statistics does not leak more information compared to FedAvg. ... I understand that it could be orthogonal to the problem under study..
>
> Thanks for pointing out the issue and here we establish the claim.
> Recall that we have the following statements about the communication  "Noticeably, AT users will share the statistic difference ($\Delta \mu$) instead of local statistics ($\mu$) with the server, and the server will send the averaged parameters with ST users only." (in the last 2nd sentence of the last paragraph of Sec 4).
> Under this setting, we consider the privacy leakage in one federated communication round.
> Both FedAvg and our method send the averaged (BN) parameters to users.
> Given the same shared information, the privacy leakage will be at the same level.
> Second, note that the number of rounds of sharing averaged BN parameters is different for ours and FedAvg.
> For FedRBN, the sharing happens only **once** (in Alg. 2 after federated training) while FedAvg shares the information **multiple rounds** during federated training.
> Therefore, our method in the worst case (for example, one federated training with FedRBN versus one-round federated training with FedAvg) shares the same amount of privacy information as the FedAvg.
> In most realistic cases, the training takes more than one communication round, and therefore our method suffers less privacy leakage.

---

> ### Author Response · Authors · 2021-11-27
> **A kind reminder**
>
> Dear Reviewer p9LS,
>
> Thank you for your time to review our paper and leave valuable comments and suggestions. As the discussion period will end soon, we are wondering whether you have got a chance to read our response to your questions? We want to make sure all your concerns are addressed on time if any.
>
> Authors

---

### Author Response · Authors · 2021-11-23
**General response and a kind reminder**

Dear all reviewers:

We want to thank all the reviewers for the constructive suggestions and thoughtful reviews, which are important for improving our paper. As a follow-up on our rebuttal, we would like to kindly remind the reviewers that the close date of the discussion is approaching. We hope to use this open response window to discuss the paper, answer follow-up questions, and improve the quality of our paper.

* We thank reviewer bKE4 for strongly supporting the originality and significance of our paper.
* We thank reviewer B9Be for the positive re-consideration after reading our rebuttals.
* We also thank reviewer dgjR for being positive with the novelty, importance of our work, and providing detailed suggestions on in-depth analysis and discussion of our work. We would like to have further discussion if any follow-up question is raised.
* We sincerely hope to have further discussion with reviewer udUC and p9LS and make sure the reviewers found our responses are convincing. We have tried our best to address the reviewers' concerns, and we can clarify more if there is more need. We are happy to answer any additional questions and provide more information.

---

### Author Response · Authors · 2021-11-29
**Summary of updates to reviewers and AC panel**

Dear reviewers and AC panel,

We appreciate all reviewers for reviewing our paper and leaving valuable comments! We are grateful that the merits of our work are acknowledged by Reviewer bKE4, B9Be, and dgjR.

With all respect, we think some comments from Reviewer p9LS and udUC are biased due to a misunderstanding of our paper. We have tried to address his/her concerns but we still have not gotten further comments from Reviewer p9LS or udUC until now (the last day of the discussion period). Due to the situation, we would like to bring the discussions to the AC's attention.

1. **Concern of novelty.** As stated by several reviewers including Reviewer p9LS (in ‘Pros’), our work ‘proposes a method to transfer robustness built on prior observation of BN correlation, decouple BN for noise and clean samples and de-bias estimation of robust BN statistics for AT-to-ST transfer.‘ (paraphrased from Reviewer p9LS’s comments). Reviewer dgjR also finds that ‘It is interesting to propagate adversarial robustness via transfer of batch-normalization statistics from high-resource users to low-resource users.’ Reviewer bKE4 mentioned that ‘The proposed method is simple and intuitive.’ However, we surprisingly find that Reviewer p9LS’s negative comments (i.e., ‘marginal contribution’) contradict not only other reviewers’ but also his/her opinions.
2. **Evaluation by stronger attacks.** In our initial submission, we have presented multiple attacks including SOTA attacks (auto-attack) and popular attacks (PGD), white-box and black-box (LSA) attacks, and joint attacks on the noise detector and models. These evaluations explore multiple dimensions of our robust methods. Regarding the challenges on the possible failure of noise detector raised by Reviewer p9LS, we want to quote Reviewer bKE4’s words as our response: ‘the evaluation is very thorough and convincing.’
3. **Using PGD in adversarial training** is a common practice that has been demonstrated effective in multiple papers [1,2,3]. Auto-attack is strong but has not been used for adversarial training so far. As a result, we disagree with Reviewer p9LS on using AA for training.
4. **Rationality of DBN** for adversarial training has been demonstrated in [4]. Though concerned by Reviewer udUC, the limitation of the DBN is observed when robust BN components are not transferred between clients instead of the affine sharing. As evidenced by our ablation study, copying robust BN components can mitigate the issue.
5. **Clarifications on technical details.** Though some reviewers find our paper is well written and easy to follow (for example, Reviewer dgjR and B9Be), Reviewer udUC thinks some technical details are not clear. We reiterate our points in our paper and tried to address these concerns by including more discussions (particularly the multi-task federated learning) in our revision. We believe Reviewer udUC may misunderstand some core settings, e.g., only partial users do adversarial training, which likely results in his/her negative opinions.

* [1] Hongyang Zhang, et al. Theoretically principled trade-off between robustness and accuracy. In ICML, 2019.
* [2] Aleksander Madry, et al., Towards deep learning models resistant to adversarial attacks. ICLR, 2018.
* [3] Tianlong Chen, et al. Adversarial robustness: From self-supervised pre-training to fine-tuning. CVPR, 2020.
* [4] Xie, C., et al. Adversarial Examples Improve Image Recognition. CVPR, 2020

---

### Decision · Program_Chairs · 2022-01-20

**Decision:**

Reject

**Comment:**

This paper considered the computational budgets of adversarial training in the context of Federated Learning and studied the propagation of adversarial robustness from affordable parties to low-resource parties. Although the authors conducted the extensive experiments to show the effectivenss of FedRBN, there are still important concerns from the reviewers,

(1) The novelty is marginal compared to FedBN, DBN and previous insights, which moves the similar framework to adversarial robustness and changed the rules, especially given the competitive ICLR. More theorectical novelty will be preferred.

(2) Many technical details are not well explained and some parts need to be improved, which make the reviewers not well convinced about FedRBN.

Given above points, I will recommend rejection and encourage the authors to improve the paper in the future.